# Dynamical response of the southwestern Laurentide Ice Sheet to rapid Bølling-Allerød warming

Sophie L. Norris[1,2*], Martin Margold[3], David J. A. Evans[4], Nigel Atkinson[5] and Duane G. Froese[2*]

[1]Department of Geography, University of Victoria, David Turpin Building, 3800 Finnerty Road, Victoria, BC, V9P 5C2, Canada
[2]Department of Earth and Atmospheric Sciences, 1-26 Earth Sciences Building, University of Alberta, Edmonton, AB, T6G 2E3, Canada
[3]Department of Physical Geography and Geoecology, Charles University, Albertov 6, 128 43 Praha 2, Czech Republic
[4]Department of Geography, Durham University, South Road, Durham, DH1 3LE, United Kingdom
[5]Alberta Geological Survey, Alberta Energy Regulator, 402 Twin Atria Building, 4 4999-98 Avenue, Edmonton, AB, T6B 2X3, Canada
*Corresponding authors

*Correspondence to*: Sophie L. Norris (sophienorris@uvic.ca) and Duane G. Froese (duane.froese@ualberta.ca)

## Abstract

The shift in climate that occurred between the Last Glacial Maximum (LGM) and the early Holocene (ca. 18-12 ka BP) displayed rates of temperature increase similar to present-day warming trends. The most rapid recorded changes in temperature occurred during the abrupt climate oscillations known as the Bølling-Allerød interstadial (14.7-12.9 ka BP) and the Younger Dryas stadial (12.9-11.7 ka BP). Reconstructing ice sheet dynamics during these climate oscillations provides the opportunity to assess long-term ice sheet evolution in reaction to a rapidly changing climate. Here, we use glacial geomorphological inversion methods (flowsets) to reconstruct the ice flow dynamics and the marginal retreat pattern of the southwestern sector of the Laurentide Ice Sheet (SWLIS). We combine our reconstruction with a recently compiled regional deglaciation chronology to depict ice flow dynamics that encompass the time period from pre-LGM to the early Holocene. Our reconstruction portrays three macroscale reorganizations in the orientation and dynamics of ice streaming followed by regional deglaciation associated with rapid warming during the Bølling-Allerød interstadial. Initial westward flow is documented, likely associated with an early set of ice streams that formed during the advance to the LGM. During the LGM ice streaming displays a dominant north to south orientation. Ice sheet thinning at ~15 ka is associated with, a macroscale reorganization in ice stream flow, with a complex of ice streams recording south-eastward flow. A second macroscale reorganization in ice flow is then observed at ~14 ka, in which southwestern ice flow is restricted to the Hay, Peace, Athabasca, and Churchill river lowlands. Rates of ice sheet retreat then slowed considerably during the Younger Dryas stadial; at this time, the ice margin was situated north of the Canadian Shield boundary and ice flow continued to be sourced from the northeast. Resulting from these changes in ice sheet dynamics, we recognize a three-part pattern of deglacial landform zonation within the SWLIS characterized by active ice margin recession, stagnation and downwasting punctuated by local surging (terrestrial ice sheet collapse): the outer deglacial zone contains large recessional moraines aligned with the direction of active ice margin retreat; the intermediate deglacial zone contains large regions of hummocky and stagnation terrain, in some areas crosscut by the signature of local surges, reflecting punctuated stagnation and downwasting; and the inner deglacial zone contains inset recessional moraines demarcating progressive regional ice margin retreat. We attribute these macroscale changes in ice flow geometry and associated deglacial behaviour to external climatic controls during the Bølling-Allerød and Younger Dryas but also recognize the role of internal (glaciological, lithological and topographic) controls in SWLIS dynamics.

# 1 Introduction

Mass loss from contemporary ice sheets and its contribution to rising global sea level has enhanced over the last three decades and is projected to continue to rise (Pattyn et al., 2018). These changes have mainly resulted from accelerated ice-flow rates and changes in surface mass balance, broadly linked to atmospheric and oceanic warming (Alley et al., 2005; Rignot and Kanagaratnam, 2006; Rignot et al., 2008, 2011; Shepherd, 2018). Changes in ice sheet dynamics are of particular concern given the potential sea level equivalent contribution of ice sheets in Greenland and Antarctica (Pattyn et al., 2018). However, with only

a few decades of observations, significant uncertainties remain in projecting an ice sheet's response to future climate system changes. Consequently, reconstructing the behaviour of palaeo-ice sheets and how it relates to past climatic change is important to predict future changes to contemporary ice sheets (Alley and Bindschadler, 2001; Rignot and Thomas, 2002; Clark et al., 2009; Carlson and Clark, 2012).

In this context, the complex and largely well-preserved landform imprint of the Late Wisconsinan Laurentide Ice Sheet (LIS) documents regional palaeoglaciological responses to rapid climate change following on from the Last Glacial Maximum (LGM) and consequently is known as an important source of information (Dyke and Prest, 1987; Clark, 1994; Dyke, 2004; Margold et al., 2015b; Stokes et al., 2016; Margold et al., 2018). In particular, geomorphic, sedimentary and geochemical investigations of parts of the southwestern LIS (SWLIS) have documented significant changes in ice flow orientation (Ross et al., 2009; Ó Cofaigh

et al., 2010; Evans et al., 2014; Atkinson et al., 2016; Margold et al., 2015b, 2018; Fig. 1). However, until recently, large differences between chronological methods have led to uncertainties surrounding the timing of these reorganizations. An updated deglacial chronology (Norris et al., 2022), comprising a synthesis of ~150 [10]Be terrestrial cosmogenic nuclide exposure, radiocarbon and infrared/optically stimulated luminescence dates, indicates that the SWLIS underwent rapid deglaciation in response to warming during the Bølling-Allerød (BA) interstadial (14.7-12.9 ka BP; Rasmussen et al., 2014) followed by a

decrease in ice sheet retreat rates and marginal stand-still during the Younger Dryas (YD) stadial. Here, we use glacial geomorphological mapping followed by the classification of individual features into flowsets to reconstruct the ice flow dynamics and the marginal retreat pattern of the SWLIS. We combine this ice dynamic reconstruction with existing chronologies to present a model of ice sheet behaviour from pre-LGM to the start of the Holocene.

Our reconstruction supports previous ideas of rapidly switching ice flow directions within the SWLIS during the last deglaciation. We propose that these ice flow changes result from three macroscale ice flow reorganizations within a ~2500 yr interval accompanied by rapid ice margin retreat and ice margin stagnation (in some areas punctuated by local surging). We attribute these macroscale changes in ice flow dynamics, in large part, to climatic changes during the BA and YD. However, we also highlight the complex interplay between this external driving mechanism and internal glaciological, lithological, and topographic controls.

# 2 Background

## 2.1 Study Area

The SWLIS covered an area of approximately 900,000 km$^2$ and was located on both the Canadian Shield and the Interior Plains. In the northeast of the region, the Canadian Shield exposes Crystalline lithologies, which in some locations are locally mantled by

a surface cover of Quaternary sediment. In the southwest of the region, the Western Canadian Sedimentary Basin overlays the Canadian Shield, and this thick package of sedimentary rock broadly slopes and thins towards the northeast (Mossop and Shetsen,

2012; Banks and Harris, 2018). In part resulting from the northeasterly thinning of the Western Canadian Sedimentary Basin, the Interior Plains exhibits a pronounced surface gradient that descends from its highest point of 1200 m a.sl. in the west to a low point of 240 m a.s.l at the transition to the Canadian Shield. A network of predominately northwest to southeasterly orientated buried preglacial valleys dissect the landscape and are eroded into the Western Canadian Sedimentary Basin (Stalker, 1961; Cummings et al., 2012). In many locations, preglacial valleys hold thick sequences made up of multiple glacial diamicts and packages of fluvial gravels and glaciofluvial silts, sands and clays (Klassen, 1989; Stalker, 1961; Evans and Campbell, 1995; Andriashek and Atkinson, 2007). Multiple uplands separate this preglacial valley system and rise up to 700 m above the gently sloping surface of the Interior Plains (Leckie, 2006).

## 2.2 Previous works

The geomorphology and sedimentology of the region encompassed by the SWLIS have been well-studied (Prest, 1968; Christiansen, 1979; Shetsen, 1984; Dyke and Prest, 1987; Evans, 2000; Dyke, 2004; Evans et al., 2008; Fisher et al., 2009; Ross et al., 2009; Ó Cofaigh et al., 2010; Kleman et al., 2010; Evans et al., 2012, 2014; Margold et al., 2015b; Atkinson et al., 2016; Utting et al., 2016; Margold et al., 2018; Evans et al., 2020, 2021). Within the region, cross-cutting landforms, as well as variation in striations, till lithologies and geochemical signatures document changes in ice flow orientation. Relatively older and overprinted southwesterly ice flows have previously been suggested to be associated with an advancing ice sheet (Ó Cofaigh et al., 2010; Margold et al., 2015b). A dominant southern flow has been attributed to LGM conditions, and a subsequent southeastward and then southwest flow was thought to have operated during ice retreat (Evans, 2000; Evans et al., 2008; Ross et al., 2009; Ó Cofaigh et al., 2010; Evans et al., 2012, 2014; Margold et al., 2015b, a, 2018; Evans et al., 2020, 2021). While the reconfiguration of these ice streams can be clearly seen in the geomorphic record, they have not been reconstructed in detail for the entire ice sheet sector. Further, the timing of changes in ice sheet dynamics and how they may relate to external climate forcing or internal forcing mechanisms (i.e., glaciological, lithological, and topographic changes) remains largely undetermined.

Until recently, the deglacial chronology of the SWLIS was based primarily on radiocarbon data (in compilations by Dyke, 2004; Dalton et al., 2020), the application of which is limited by the unknown temporal offset between deglaciation and biotic colonization (c.f. Froese et al., 2019). Furthermore, where infrared/optically stimulated luminescence dating and a small number of [10]Be terrestrial cosmogenic nuclide exposure ages have been used to complement radiocarbon chronologies, significant differences between dating methods have meant that our understanding of the deglacial response of the SWLIS to climate drivers has remained limited. Norris et al. (2022) recently addressed these issues by combining existing luminescence and the most viable minimum-limiting radiocarbon dates with new [10]Be exposure ages to establish the timing of SWLIS deglaciation. Within a Bayesian framework, this compilation unified the ages provided by these methods into an internally consistent chronology, which indicates that the SWLIS retreated from its southernmost coalescence with the CIS to its YD position along the Cree Lake Moraine (Fig. 1) in ≤ 2500 yrs (Norris et al., 2022).

## 3 Methods

### 3.1 Geomorphological mapping

In order to provide a complete reconstruction of the ice flow configuration and dynamics as well as the ice-marginal retreat, the glacial geomorphology of the SWLIS was collated from pre-existing mapping (Ó Cofaigh et al., 2010; Atkinson et al., 2014; Evans

et al., 2016; Norris et al., 2017; Atkinson et al., 2018; Evans et al., 2020). The following landform categories were collated: ice flow parallel lineations (drumlinoid ridges, mega-scale glacial lineations (MSGL), and crag-and-tail features), moraine crests

(lateral shear, terminal, and recessional), ice-thrust ridges (and glaciotectonic rafts), crevasse fill ridges, meltwater channels and eskers, kames/hummocks, palaeo-shorelines, dunes, and undifferentiated ridges. Pre-existing mapping comprises features that were predominantly identified from satellite imagery combined with detailed aerial photograph interpretation. In many locations, features have been ground-truthed by the original authors (e.g., Matthews et al. 1975; Atkinson et al. 2014, 2018 and references therein). In particular, geomorphic mapping from LiDAR imagery (5-15 m resolution) at a 1:10,000 scale was utilized for Alberta

(Atkinson et al., 2014, 2018). For all other parts of the study area, geomorphic mapping was compiled at 1:30,000 (Norris et al., 2017), 1:10,000 (Ó Cofaigh et al., 2010; Evans et al., 2016, 2020) and 1:1,000,000 (Mathews et al., 1975) scales. Resulting from an absence of LiDAR data within Saskatchewan and northeast British Columbia, mapping has been primarily completed from a combination of 1-arc second SRTM (~30 m resolution), ALOS (~30 m resolution) imagery and aerial photograph analysis.

### 3.2 Glacial inversion (flowset classification)

We utilized a glacial geomorphological inversion method or flowset approach (see Kleman and Borgström, 1996; Kleman et al., 1997, 2006; Greenwood and Clark, 2009), which formalizes standardized geomorphological mapping techniques (Dyke et al. 1992). The approach applies our pre-existing knowledge of landform generation to allow us to reconstruct ice sheet properties using features formed at the bed of a palaeo-ice sheet (Fig. 2) (Kleman et al., 1997, 2006). Ice flow patterns were grouped into flowlines and then flowsets based on their morphometry and proximity to other glacial landforms (Clark, 1999). The degree of

orientation similarity in lineation was also taken into account, following the assumption that all lineations in the same flowset should have similar orientations (Clark, 1999). Following the procedures outlined by Atkinson et al. (2016), glacial inversion methods were combined with a landsystems approach (see Evans, 2007, 2013). This enables landforms and their constituent sediment to be combined into assemblages related to genetic processes, which facilitates a more comprehensive evaluation of 'process form' to be included in the assessment of palaeo-ice sheet and ice streaming dynamics (Evans et al., 2008, 2014; Atkinson

et al., 2016; Fig. 2). To make determinations about the sediments that comprise landforms, we utilized previously published surficial geology mapping (Simpson, 1997; Fenton et al., 2013; Geological Survey of Canada, 2014; Bednarski, 2000; Hickin and Fournier, 2011).

We synthesized and updated flowsets (mapped and originally presented by Evans et al., 2014; Atkinson et al., 2016; Utting et al.,

2016; Norris, 2019; Atkinson and Utting, 2021) and generated new flowset mapping in regions where no mapping was available (~25% of the flowsets) (Fig. 3a). In addition to mapped glacial geomorphology, in many cases, previous regional scale delineation of ice flow corridors or ice stream locations guided our flowset identification (Evans, 2000; Ross et al., 2009; Evans et al., 2008; Ó Cofaigh et al., 2010; Evans et al., 2012; Margold et al., 2015b, a; Stokes et al., 2016; Margold et al., 2018; Evans et al., 2020) (Fig. 3a). Each flowset was then divided into one of six categories: (i) ice stream, (ii) event, (iii) deglacial retreat, (iv) deglacial

readvance, (v) palimpsest, and (vi) unknown (Table 1; categories adapted from Kleman, 2006). Flowsets were then ordered according to their relative age based on their cross-cutting relations (Kleman et al., 2006; Greenwood et al., 2007; Greenwood and Clark, 2009; Kleman et al., 2010; Hughes et al., 2014) to reconstruct dynamics changes in ice flow over time.

Consistent with previous studies within the Canadian Prairies (e.g., Ross et al., 2009; Ó Cofaigh et al., 2010; Margold et al., 2015b;

Stokes et al., 2016), we refer to an 'ice stream' as a corridor within an ice sheet that is flowing relatively faster than the ice surrounding it (Paterson, 1994). We also document switches between periods of rapid ice flow and slower ice flow likely driven

by internal forcing mechanisms; in such cases, we refer to them as exhibiting 'surge' behaviour (Evans et al., 1999; Clayton et al., 1985; Evans et al., 2020) and classify them as deglacial readvance flowsets.

## 3.3 Deglacial dynamics

We place our reconstruction into a chronological context using the ice margin chronology of Norris et al. (2022). This dataset comprises ~150 [10]Be terrestrial cosmogenic exposure ages, infrared/optically stimulated luminescence, and high-quality, minimum-limiting radiocarbon ages. These data were combined within a Bayesian statistical framework to present an internally consistent retreat chronology. We combine this chronology with the mapped ice margin positions of Dyke (2004) and Dalton et al. (2020), reassigning a new age to each position based on the chronology of Norris et al. (2022), and present recessional isochrones for the SWLIS.

## 4 Results

We identify 425 flowsets in the SWLIS landform record (Fig. 3b; to view individual flowsets discussed below, see the supplemental .kml file in Google Earth). We present flowsets and their constituent landforms, which have been identified from previous studies (mapped and originally presented by Evans et al., 2014; Atkinson et al., 2016; Utting et al., 2016; Norris, 2019; Atkinson and Utting, 2021) and delineate flowsets from geomorphological mapping where none existed (Fig. 3a). All flowsets are categorized as either (i) ice stream, (ii) event, (iii) deglacial retreat, (iv) deglacial readvance, (v) palimpsest, and (vi) unknown (Table 1) and differentiated based on relative age.

Of the 425 flowsets mapped, we categorize ~55% of flowsets ice stream, 11% as event, 13% as deglacial retreat, 9% as deglacial readvance, and 8% as palimpsest flowsets. About 20 flowsets (4%) contain only a small number of landforms that are fragmented or whose origins are unclear, we categorize these flowsets as having an unknown origin. Recorded by the cross-cutting relationships of flowsets, it is clear that ice flow patterns within the SWLIS document several macroscale reorganizations (Fig. 3b). Below, we present the evolution of ice flow in five phases. While we differentiate our reconstruction into five major phases, we note that in many cases, flowsets document a time-transgressive bedform generation (see Greenwood and Clark, 2009) and reflect a rapidly fluctuating ice flow regime. As such, multiple flowsets may depict ice flow in a small region during a single phase. In these cases, the relative timing of flowset evolution is described below and documented in Figure 4.

### 4.1 Phase 1

Phase 1 documents ice flow from the northeast to the southwest. It comprises 25 flowsets (Fig. 4a) composed of drumlins and flutings. These flowsets are not associated with any ice-marginal landforms. Their imprint is mainly preserved in areas of high ground. They occur as topographically unconstrained southwestward-oriented fast-flow corridors extending for up to 60 km across the Cameron Hills (fs 417), Birch Mountains (fs 267), Clear Hills (fs 300, 348, 349, 353, 355, 358, 362, 369, 379, 388, 390), Stony Mountain (fs 123, 130, 157 and 181), Buffalo Head Hills (fs 330), and Swan Hills (fs 245). Within the eastern portions of the study area, fs 10 and 6 record southwestward flow restricted to the valley base, northeast, and southwest of Saskatoon and fs 134 and 141 record southwestward flow northeast of Edmonton. Sourced from the Rocky Mountains, northwest-orientated ice flow is recorded within the Grande Prairie region (fs 310, 324, 361).

**4.2 Phase 2**

Phase 2 documents ice flow from north to south (Fig. 4b) in multiple corridors. Within the western portions of the study area during

this phase, fs 191, 197, 198, 200, 209, 212, 214, 219, 220, 223, 278, 281 document a series of tributary flows that coalesce in the Athabasca River lowlands as a single flowpath (fs 207). These flowsets comprise a high density of MSGL as well as drumlins and flutings. This single major north-to-south aligned flow separates immediately north of Edmonton. It can be traced over 450 km as an eastern corridor (fs 38,51,93,142) and western corridor (fs 39, 57, 58, 76, 104, 111, 119, 121, 129, 148, 152, 151, 153, 160, 167, 170, 172, 186, 189, 192, 201, 204, 208) both of which end in extensive zones ice marginal landforms (Lethbridge moraine complex;

Stalker, 1962, 1977; Eyles et al., 1999; Evans et al., 2008, 2012, 2014). Cordilleran ice draining to the southeast from the Rocky Mountains is documented by fs 55, 59, 68, 70, 75, 80, 84, 87, 90, 91, 168, 170, 174, 190, 213, 242, 248, 262, 265.

Within Saskatchewan, north-to-south oriented ice flow is first documented as a small flowset (fs 137) composed of drumlins located at the southwest boundary of the Canadian Shield. Flow can then be traced to a region of widespread southwest-trending

MSGL (fs 42, 54, 61, 62, 67). This flow path then extends along the Alberta-Saskatchewan border as a succession of seven flowsets (fs 12, 16, 18, 19, 24, 30, 42), ending at a series of stacked moraines on the northern side of the Cypress Hills. In several locations, lineations comprising these flowsets are overprinted by concentrations of crevasses fill ridges. South of the Cypress Hills, fs 5, 7, 8 document evidence of small ice flows that extend around this upland to the south.

**4.3 Phase 3**

Phase 3 documents ice flow from the northwest to southeast (Fig. 4c). Large regions of streamlined subglacial bedforms occur over six broad corridors. These corridors separate the widespread hummocky terrain (in some places locally overprinted by surge-type geomorphic signatures) that characterizes much of the central SWLIS. In several locations, these streamlined zones are overprinted by swaths of crevasses fill ridges that are oriented perpendicular to the ice flow direction. In addition, the boundary of

these fast-flow corridors is often demarcated by 'ice stream or lateral shear moraines' (see Ross et al., 2009; Ó Cofaigh et al., 2010).

Within the northern portions of the Churchill River lowlands, fs 26, 34, 40, 85, 149,150 record ice flow that cross-cuts phase 2 southerly oriented flow. The flow direction of bedforms that comprise these flowsets is difficult to identify because they are

overlain by glaciolacustrine material. However, multiple ice thrust ridges that are accompanied by small up-ice hollows (hill-hole pairs) throughout this region indicate ice flow to the southeast. Within the southern portions of the Churchill River lowlands, fs 4, 52, 81, 97, 106, 112, 124, 127, 128, 136, 145, 154, 158, 163, 166, 177, 178 record a similar ice flow that originated northwest of Cold Lake. Areas north and south of the Neutral Hills display similar southeastward-oriented flow as documented by fs 28 and 43. This ice flow was subsequently cross-cut by a slightly later south-southeast ice flow (fs 2, 3, 9, 25 and 66).


Closer to the western ice margin at this time, ice flow fed lobes in the Calgary (fs 72, 74, 82, 92, 99, 102, 116, 121, 131, 144, 148, 151, 160, 162, 167, 172, 179, 180, 192, 193, 202, 208) and Lethbridge regions (fs 11, 13, 14, 15, 21, 22, 23, 27, 29, 31, 35, 38, 41, 44, 46, 47, 50, 51, 53, 93, 95, 100, 133, 138, 147), recording southwestward ice flow orientations. Within the Grande Prairie region,

small ice lobes were fed by Cordilleran (fs 213, 265, 272, 354) and Laurentide ice (fs 250, 280, 288, 291, 296, 304, 305, 314, 320, 341, 342, 363). In several areas, overprinting of these wide ice flow corridors occurs. These are best documented surrounding the Neutral Hills, where flowsets appear to have been sourced from the upland and overprint parts of the regional scale southwestward flows within the surrounding lowlands (e.g., fs 32, 33 48, 60, 79, 88, 98, 110). In several cases, these flowset overprint or are composed of glaciolacustrine material, suggesting that proglacial lakes were present in the landscape during their formation.

**4.4 Phase 4**

Phase 4 documents southwestward oriented ice flow in the Hay (e.g., fs 384, 396, 398, 419, 425), Peace (e.g., fs 308, 317,326 and 335) and Athabasca (e.g., fs 211, 214, 219, 222, 227, 236, 244, 254, 255) river lowlands (Fig. 4d). The geomorphic imprint of southwesterly ice flow is also visible on the upper surface of the Birch Mountains (fs 253 and 264). Within the Churchill River lowlands, a southeastward flow is recorded (fs 25, 49, 77, 83, 125, 146) (Fig. 4d). On a local scale, several small-scale shifts in ice flow direction are documented during this phase. Firstly, associated with fs 150, the cross-cutting relationship of two well-preserved lateral shear moraines reveals a progressive 'clockwise' migration of ice flows. This cross-cutting pattern likely signifies a shift in the ice drainage migrating from a west-to-north source as the ice sheet thinned over the Stony Mountains (i.e., from Phase 3 to 4). Secondly, fs 146 and 125 document another local ice flow migration. These flowsets document a shift from southwestward ice streaming to northwestward ice flow (feeding flows within the Hay and Peace River ice flows). Glacial lineations in these flowsets blend with one another, implying a migration rather than a simple overprinting (see also Margold et al., 2015a, b, 2018). It should also be noted that a similar time transgressive 'clockwise' migration of ice flow and lateral shear moraines is also observed immediately east of the SWLIS by Ross et al. (2009) in the region south of Prince Albert, Saskatchewan.

**4.5 Phase 5**

Phase 5 documents the last stages of ice flow to the southwest (Fig. 4e). During this phase, ice flow terminated at the Cree Lake Moraine and Upper Cree Lake Moraine complex. A large regional scale flowset (fs 164) and a series of smaller retreat and localized readvance flowsets (fs 65, 77, 94, 101, 114, 118, 135, 140, 161) cover areas of the exposed crystalline bedrock of the Canadian Shield as well as regions covered by patches of Quaternary sediment.

In addition to flowsets associated with regional deglaciation, this phase documents ice draining radially from several uplands. Flowsets on the Cameron Hills (fs 418, 424), Caribou Mountains (fs 332, 339, 352, 356), Birch Mountains (fs 260 and 275), Clear Hills (fs 368, 373, 376), and Swan Hills (fs 274) comprise suites of lineations and ice-marginal channels, eskers, and recessional moraines that overprint flowsets documenting regional deglaciation (phase 4 flowsets). These landforms also cross-cut recessional moraines and glaciolacustrine sediments within the surrounding lowlands, suggesting that the lowland regions were ice-free and that proglacial lakes in the area had drained (see also, Atkinson et al., 2016).

**5 Dynamic evolution of the SWLIS**

To place our ice sheet reconstruction into chronological context, we integrate our flowset and ice-marginal geomorphologic mapping with the deglacial chronological synthesis of Norris et al. (2022). We combine this ice margin chronology with the mapped ice margin positions of Dyke (2004) and Dalton et al. (2020), reassigning a new age to each position based on the chronology of

Norris et al. (2022), and reconstruct ice sheet dynamics and configuration at five phases ranging from pre-LGM to the early Holocene (Fig. 4).

**5.1 Pre-LGM ice flow pattern (Phase 1)**

Phase 1 flowsets likely document the advance towards convergence of the CIS and LIS before the LGM. In many cases, flowsets relating to this phase are small and fragmented. To avoid overinterpretation of the geomorphic record, we do not delineate ice stream pathways associated with the pre-LGM ice flow pattern, however in regions where these have previously been identified, we include them in our interpretation (see Fig. 4). Evidence for southwestward fast flow relating to this phase is largely restricted to upland areas (Buffalo Head, Cameron, Clear and Swan Hills and Birch and Stony Mountains) within the central portion of the SWLIS (Fig. 4a). Flowsets on the Cameron Hills are broadly consistent with a region of ice streaming highlighted by Margold et al. (2015a, b) and are interpreted as their 'Cameron Hills Fragments'. Within the eastern part of the study area, southwestward flow is also preserved in the lowland regions surrounding Saskatoon and northeast of Edmonton. Portions of our flowsets in these regions have previously been noted by Ó Cofaigh et al. (2010), and Margold et al. (2015a, b). We interpret our flowsets as an extension of their 'Pre 1 Fragments' and 'Winefred Lake Fragments', respectively. Initial convergence between the LIS and CIS is recorded in the Grande Prairie region. These flows were originally delineated into flowsets by Atkinson et al. (2016) and depict that initial eastward flows have been deflected northwestward, likely by the LIS. Subglacial landforms that comprise flowsets in the Grande Prairie and Calgary regions have been interpreted by Atkinson et al. (2016) to signify synchronous build-up of the CIS and LIS. We support this interpretation and suggest that these flowsets must have been active prior to the formation of an extensive CIS-LIS ice saddle, which would have deflected ice flow to the south.

Within southern Alberta, we do not document any flowsets relating to this phase, although 'overridden glaciotectonic composite ridges' have been mapped previously in the region (see Evans et al., 2008, 2012, 2014). These landforms demarcate the lobate margins of what would become major LGM ice stream tracks (see Evans et al., 2008, 2012, 2014). Minimal chronology relating to the pre-LGM ice sheet evolution of the SWLIS is available. However, the ice flow orientation documented here is consistent with the ice flow and ice margin advance direct shown by previous continental-scale reconstructions (Dyke et al., 2002; Batchelor et al., 2019; Dalton et al., 2022) during the transition to the local LGM. This interpretation also agrees with till geochemical analyses that show calcium oxide concentrations to be elevated in tills across southeastern Alberta, which have been linked to a dolomite boulder dispersal train extending over >100 km from central Saskatchewan (Westgate, 1968; Pawluk and Bayrock, 1969).

**5.2 LGM ice flow pattern (Phase 2)**

During the LGM, flowsets record the development of three north to south-oriented ice streams. This ice flow orientation is broadly consistent with previous regional geomorphic reconstructions (Ross et al., 2009; Ó Cofaigh et al., 2010; Margold et al., 2015b, a, 2018; Evans et al., 2008, 2014) (Fig. 4b) and supports the notion of an extensive ice dome and ice (divide) saddle situated to the north (Dyke and Prest, 1987; Margold et al., 2015b, 2018; Tarasov et al., 2012). Within the eastern portion of the SWLIS, we document the flowpath of the region's most longitudinally extensive ice stream. This ice stream flowpath was first mapped by Prest et al. (1968) and formally identified as the East Lobe by Shetsen (1984) and later delineated by Ross et al. (2009) as the Maskwa Corridor and by Ó Cofaigh et al. (2010) as Ice Stream 1. To avoid confusion, we herein refer to this flowpath as the Maskwa Ice Stream (Maskwa IS). The onset zone of the Maskwa IS is cross-cut by later ice flows and is only preserved as a single flowset (fs 137). We suggest this flowset documents the southern portion of the convergent flow that sat north of the Canadian

Shield boundary. An onset zone near the Canadian Shield boundary is also supported by regional airborne gamma-ray spectrometry surveys (GSC, 2008; Ross et al., 2009), which document the distribution of potassium-rich glacial diamict transported from the Canadian Shield along a north-to-south flowpath that is broadly aligned with fs 137.

The Maskwa IS extended from its onset zone and ended on the northern side of the Cypress Hills (after Ross et al., 2009; Ó Cofaigh et al., 2010; Margold et al., 2015b, 2018). The geomorphic imprint of the Maskwa IS on the Cypress Hills likely correlates to the highest elevation of the Elkwater drift recorded on this upland (see Westgate, 1968; Evans et al., 2014). Transporting ice flow south, the Maskwa IS fed two small ice lobes that flowed around the east and west of the Cypress Hills (herein named East and West Cypress Hills fragments) (Klassen, 1994; Ross et al., 2009; Ó Cofaigh et al., 2010; Evans et al., 2016).

Within the Athabasca River lowlands, we document the onset zone of another north-to-south oriented ice stream. Due to its association with previous work further to the south (Prest et al., 1968; Evans et al., 2008; Atkinson et al., 2014), we interpret this as the onset zone of the Central Alberta Ice Stream and High Plains Ice Stream (hereafter CAIS and HPIS) (Prest et al., 1968; Evans, 2000; Evans et al., 2008; Margold et al., 2015a, b). These ice streams flowed as a single pathway until they separated immediately northeast of Edmonton, the western arm becoming the HPIS. The HPIS then merged with ice flowing from the CIS to form the Rocky Mountain Foothills Ice Stream (Evans et al., 2008, 2012, 2014; Utting et al., 2016; Margold et al., 2015b, 2019). The onset zone of the CAIS/HPIS in the Athabasca River lowlands region does not display a zone of convergent flow (i.e. Prest et al., 1968; De Angelis and Kleman, 2008). While it is possible that the zone of convergence for these ice streams may not have been preserved in the geomorphic record (e.g., Evans et al., 2014) we instead suggest that it did not have a convergent onset zone while active, but rather was sourced by multiple smaller feeder tributaries. This is justified by the small changes in ice stream orientation recorded within the Athabasca River lowlands and the existence of multiple flowsets (e.g., fs 198, 214, 261, 278 and 281), which feed into this region.

In line with previous reconstructions, we suggest that the SWLIS would have drained via the Maskwa IS, CAIS, HPIS, and Rocky Mountains Foothills IS following the development of the Keewatin Ice Dome and CIS-LIS saddle and thus likely the maximum buildup of the coalescence zone between the CIS-LIS (Dyke and Prest, 1987; Margold et al., 2015b, 2018). While the age of ice stream initiation is not well known, below till radiocarbon dates in central Alberta indicate that they must postdate the incursion of the SWLIS into the region after ~25 ka BP (Young et al., 1994).

In contrast to other sectors of the LIS, LGM ice streaming in the SWLIS flowed unconstrained by topography (Ross et al., 2009; Ó Cofaigh et al., 2010; Margold et al., 2015b, 2018). This has been suggested by previous studies to result from the large ice thicknesses over the SWLIS due to its proximity to the southwestern portion of the Keewatin Ice Dome (Ross et al., 2009; Ó Cofaigh et al., 2010; Margold et al., 2015b, 2018). Additionally, unlike other sectors of the LIS, the lack of large troughs or valleys throughout the region would have meant the SWLIS was predisposed to having ice streams unconstrained by topography in the presence of extensive ice thickness (Ross et al., 2009; Ó Cofaigh et al., 2010; Margold et al., 2015b, 2018). Numerical models estimate average LGM ice thicknesses in these regions vary from >1500 to <2000 m (Tarasov et al., 2012; Peltier et al., 2015; Gowan et al., 2016; Lambeck et al., 2017). Similar ice thicknesses are also evidenced by observations of glacial isostatic adjustment observed through mapping of postglacially delevelled lake shorelines (e.g., glacial lakes Agassiz and McConnell) (Breckenridge, 2015).

**5.3 Rapid downwasting of the LIS and CIS convergence zone (Phase 3)**

We map 103 flowsets associated with this ice flow phase that documents a shift from north to south-oriented LGM flows to southeasterly flows (Fig. 4c). Many of these flows have been mapped and interpreted previously as the product of a macroscale switch in the ice dynamics associated with rapid ice sheet thinning and downwasting of the CIS-LIS convergence zone (Ross et al., 2009; Ó Cofaigh et al., 2010; Margold et al., 2018). Southeastward ice flows within the SWLIS were first highlighted by Ross et al. (2009) and Ó Cofaigh et al. (2010), who documented IS2 (Buffalo Corridor of Ross et al., 2009) that cross-cut the Maskwa

IS. We document these ice streams and map similar southeastward-oriented flows northeast of the Mostoos Hills and north of the Cypress Hills (we herein refer to this collection of ice streams as the IS2 complex) (Fig. 4c).

The change from the topographically unconstrained system of the Maskwa IS, CAIS and HPIS to the topographically constrained southeastward flow of the IS2 complex has been proposed to reflect CIS-LIS saddle collapse and rapid ice downwasting centred in the CIS-LIS convergence zone (Fig. 4c) (Margold et al., 2018). Thinning of the CIS-LIS convergence zone is constrained by

[10]Be terrestrial cosmogenic exposure dating from the northwest LIS, indicating that the region underwent rapid surface thinning starting at ~14.9-14.3 ka BP (Stoker et al., 2022). In addition, numerical model simulations (Tarasov et al., 2012; Gregoire et al., 2016) suggest that thinning and initial separation of the CIS and LIS occurred extremely rapidly, broadly correlated with the start of BA warming (ca. 15 ka BP) (Gregoire et al., 2016; Stoker et al., 2022). We note, however, that during this ice flow phase, an ice divide extending from the Keewatin Ice Dome to the west must have remained (at least in part) present to support flow in a

southeastward direction (see also Ross et al., 2009; Margold et al., 2015b, 2018). Due to substantial ice thinning, this ice divide would no longer have been fed by ice draining across the Rocky Mountains (Margold et al., 2018), so we no longer refer to it as the CIS-LIS saddle. Instead, this divide would have been situated broadly over the Caribou Mountains in northern Alberta (precise position undefined) and is herein referred to as the Plains Ice Divide (after Dyke and Prest, 1987).

Following the initial separation of the LIS and CIS, the SWLIS experienced rapid downwasting and retreat (Norris et al., 2022). Comparing the initial position of multiple south-eastward flowing networks, we agree with the previous suggestions of Ross et al. (2009), Ó Cofaigh et al. (2010) and Margold et al. (2015b, 2018), who proposed that these ice streams would have stopped after the development of a deglacial corridor, commonly referred to as the 'ice-free corridor', between the CIS-LIS; otherwise, the ice streams would have lacked an accumulation zone to drive flow southeastward. Norris et al. (2022) suggest the timing of LIS retreat

across the Interior Plains also implies that following initial detachment from the CIS, rapid retreat rates (380-340 m/yr) continued to characterize SWLIS deglaciation, contemporaneous with BA warming. This chronology suggests that southeasterly-oriented ice flows (IS2 complex) would have been very short-lived, operating for as little as ~500 yrs before they stopped and westerly flows became active in their place as the ice margin retreated.

**5.4 Deglaciation of the Interior Plains (Phases 4 and 5)**

**5.4.1 CIS-LIS separation**

Our flowset mapping, in combination with previous detailed geomorphic mapping by Evans et al. (2014) suggests that concurrent with initial separation, the lobate margins of the HPIS, CAIS, and Maskwa IS experienced a predominantly active recession, as documented by the Frank Lake, Lethbridge, and Cypress Hills moraine complexes, respectively (Fig. 4d). Zones of linear hummocky moraine (controlled moraine) and push moraine ridges that comprise these ice-marginal complexes suggest that the

ice-stream margin underwent repeated switches between polythermal and warm-based conditions during ice retreat (Evans et al.,

2014). North of the Lethbridge moraine, flowsets that mark the recession of the HPIS are, in some places, partially overprinted by ice thrust ridges ('thrust moraines' of Evans et al., 2014) and a secondary set of lineations sourced from the CAIS; this landform assemblage supports the previous detailed geomorphic mapping and sedimentological observations of Evans et al. (2008, 2014) of a later-stage CAIS fast flow event in this region (Fig. 4d). During the separation of these two ice lobes regional drainage would have been blocked, which would have resulted in multiple small ice-dammed glacial lakes forming at the margin of the HPIS (e.g., glacial lake Cardston, see Bretz, 1943; Horberg, 1952; Utting and Atkinson, 2019) and the CAIS (e.g., glacial lake Beiseker and Bassano; see Utting and Atkinson, 2019).

The timing of the initial separation of the CIS and the LIS is constrained based on a synthesis of [10]Be terrestrial cosmogenic nuclide exposure, radiocarbon, and infrared/optically stimulated luminescence dates that span the Alberta-British Columbia border zone (Norris et al., 2022). This chronology suggests that the SWLIS's initial separation began at ~14.9 ka BP (Margold et al., 2019; Norris et al., 2022). Interestingly, [10]Be exposure ages from additional sites in northwestern Alberta and northeastern British Columbia suggest slightly later separation of the two ice sheets at ~13.8 ka BP (Clark et al., 2022b). We do not adjust our timing of initial CIS-LIS separation based on these additional sites but rather continue to use an age of ~14.9 ka BP because this age is supported by three different geochronological techniques displaying internally consistent results (Norris et al., 2022). However, we emphasize that the inclusion of these additional [10]Be exposure sites from Clark et al. (2022b) would imply that the deglaciation of the SWLIS occurred even more rapidly than previously thought, and thus, the ice dynamical switches discussed below would span an even shorter timescale.

**5.4.2 Deglaciation of central Alberta and Saskatchewan**
The geomorphic record in the central portions of Alberta and Saskatchewan (i.e., northeast of Calgary) suggests deglaciation of the region was the result of widespread stagnation in places punctuated by localized surging (Fig. 5). Within the ice stream trunk zones of the CAIS, Maskwa IS, and IS2 and 3, ice sheet stagnation is demarcated by regions of lineations (MSGL, drumlins, and flutes) which are overprinted by crevasse fill ridges (Ó Cofaigh et al., 2010; Evans et al., 2016). Preservation of crevasse fill ridges during active ice recession is very uncommon; therefore, their presence in central Alberta and Saskatchewan indicates that ice in this region likely downwasted in place rather than undergoing step-back recession (Sharp, 1985; Evans, 2007; Evans et al., 2016). Bordering the flowpaths of the ice streams described above, large zones of hummocky terrain cover the landscape. Consistent with the work by Ross et al. (2009) and Ó Cofaigh et al. (2010), we suggest that ice stagnation here resulted from ice stream switching and the scavenging of the central portions of the CAIS and Maskwa IS from their accumulation zones by evolving these southeastward ice flows (IS2 complex). The short-lived southeastward flows of the IS2 complex would also have ceased following CIS-LIS separation (Ross et al., 2009; Margold et al., 2018). Thus, the ice margin in this region would no longer have been nourished, and ice stagnation would have been widespread.

In some locations flowsets comprised of ice thrust ridges, moraines, lineations, and hummocky terrain locally cross-cut large palaeo-ice stream footprints (Bretz, 1943; Evans, 2000; Evans et al., 2008, 2014, 2016, 2020; Ó Cofaigh et al., 2010). This is particularly evident in central Alberta, at the termini of fs 100, 138, 147 (northeast of Red Deer) and the region surrounding the Neutral Hills demarcated by fs 33, 48, 60 (Evans et al., 2008; Atkinson et al., 2018; Utting and Atkinson, 2019; Evans et al., 2020). These landforms confirm the development of small surging ice lobes often sourced from upland regions during ice sheet stagnation. In agreement with previous work, we suggest this behaviour was most likely conditioned by the increased availability of subglacial meltwater and/ or the expansion of local proglacial lakes around areas that contained downwasting ice (Evans et al., 2008, 2020;

Utting and Atkinson, 2019). Radiocarbon dates suggest the development of biologically viable conditions over the central SWLIS by ~13.3 ka BP (e.g., Edmonton region; Heintzman et al., 2016; Norris et al., 2022). [10]Be exposure, luminescence, and radiocarbon dates indicate that there were ice-free conditions as far north as the city of Cold Lake by ~13.5 ka BP (Norris et al., 2022). We, therefore, suggest that deglaciation, predominantly via large-scale ice sheet stagnation and downwasting in central Alberta and

Saskatchewan occurred rapidly at 13.5 ka. However, we note that portions of the landscape may have remained 'ice-cored' for much longer (see section 6.3.2).

### 5.4.3 Deglaciation of northern Alberta and Saskatchewan

Within the Churchill, Athabasca, Hay, and northern portions of the Peace river lowlands, we observe widespread active retreat of

the ice margin, fed by southwestward ice streaming of the Hay, Peace, and Athabasca ice streams (after Margold et al., 2015b) and southeastward ice streaming of Ice Stream 4/5 (IS4/5) (after Ó Cofaigh et al., 2010) (Fig 4d). We also document small time-transgressive changes in the direction of these ice streams (e.g., northeast of the Caribou Mountains feeding the Buffalo Ice Stream of Margold et al. (2015b, 2018) and on the Stony Mountains feeding IS4/5). As noted by Margold et al. (2015b), these ice streams were initially unconstrained by regional topography, as evidenced by ice stream flow over the Birch Mountains. However, our

flowset mapping and the previous large-scale reconstructions of Margold et al. (2018) suggest these ice streams were gradually restricted to the southwestward-oriented valley systems due to ice sheet retreat and thinning. This transition is marked by a series of overprinted flowsets in the Athabasca River lowlands that document the time-transgressive alteration and increasing influence of topographic controls as the ice sheet thinned.

Within all four lowlands, arcuate moraines record ice-marginal recession as a series of small lobes (Fig. 4e; Christiansen, 1979; Fisher et al., 2009). Ice retreat in these lowlands led to the development of several large glacial lakes (Fig. 6). Within the Peace River lowlands, the Peace and Wabasca lakes occupied the region immediately in front of the ice margin on the north and south sides of the Buffalo Head Hills, respectively (Utting and Atkinson, 2019). In the Athabasca River lowlands, glacial Lake Algar was the first lake to occupy the region (Utting and Atkinson, 2019). Following progressive ice retreat in the Athabasca and

Churchill river lowlands, Glacial Lake McMurray/Meadow occupied the region, and the Beaver River Moraine demarcates the position of the ice margin at this time (Fig. 6; Christiansen, 1979; Fisher et al., 2009; Anderson, 2012; Utting and Atkinson, 2019). Glacial Lake McMurray/Meadow would have continued to expand until the ice margin was far enough to the northeast to facilitate drainage either into Glacial Lake Agassiz to the southeast (see below) or Glacial Lake McConnell to the north (Fig. 6; Lemmen et al., 1994; Fisher et al., 2009; Utting and Atkinson, 2019).


[10]Be terrestrial cosmogenic exposure dating suggests that the northern portion of the Athabasca River lowlands (the region surrounding the Clearwater Lower Athabasca Spillway, see Fig. 4d for location) was ice-free by ~13.0 ka BP (Norris et al., 2022). This supports luminescence chronologies from the same region that encapsulate a broader but consistent timing of deglaciation (Munyikwa et al., 2017, 2011; Woywitka, 2019).  In contrast, radiocarbon dates within the Athabasca River lowlands indicate

deglaciation occurred almost 3 ka later, at ~10.7-10.3 ka BP (Dyke, 2004; Fisher et al., 2009; Dalton et al., 2020; Norris et al., 2022). However, compared to older, luminescence and radiocarbon dates to the south, these samples provide only minimum-limiting ages for biological productivity in the region (Froese et al., 2019; Norris et al., 2022). In contrast to the Athabasca River lowlands, there are limited chronological constraints within the northern portions of the Peace and Churchill river lowlands. However, based on the geometry of ice-marginal landforms, ice streaming would have been largely synchronous in all three of

these lowlands.

### 5.4.4 Younger Dryas ice flow pattern and deglaciation

The YD position of the SWLIS is demarcated by the Cree Lake Moraine and Upper Cree Lake Moraine, located immediately up-ice of the Canadian Shield boundary (Fig. 4e). [10]Be terrestrial cosmogenic exposure dating from two sites located on the crest of the Cree Lake Moraine and Upper Cree Lake Moraine (herein Cree Lake Moraine complex) indicate the region was ice-free at ~12.7 ka BP and 12.3 ka BP, respectively (Norris et al., 2022). Concurrent with ice retreat to the Cree Lake Moraine complex position, Glacial Lake Agassiz occupied the Churchill River lowlands to a maximum elevation recorded by shorelines (Fisher and Smith, 1994; Murton et al., 2010; Breckenridge, 2015; Young et al., 2021). Ice sheet stabilization at the Cree Lake Moraine complex would have permitted drainage of Glacial Lake Agassiz through the Clearwater Lower Athabasca Spillway (Fig. 6) to the northwest at some point during the YD stadial (Norris et al., 2021). [10]Be terrestrial cosmogenic exposure dating <50 km northeast of the Cree Lake Moraine complex document deglaciation by 11.2 ka BP, suggesting a stillstand of ~1 ka during the YD. Unlike the lobate, topographically constrained ice flows that occurred in the Churchill, Athabasca, and Peace river lowlands as the ice margin retreated onto the Canadian Shield, it was fed by a regionally extensive fast ice flow event that spread across the region as a broad sheet flow (Fig. 4d) (Margold et al., 2015b, 2018).

In addition to ice flow on the Canadian Shield, we recognize evidence of ice caps that persisted after regional deglaciation on the Birch and Caribou Mountains and Cameron and Clear Hills (Fig. 4e).  Similar evidence is also documented on the Swan Hills, previously identified by Atkinson (2009) and Atkinson et al. (2016) who document ice flow on the upland after glacial lake drainage.  In these five locations, moraines and meltwater channels that document flow into the surrounding lowlands cross-cut older flowsets as well as glaciolacustrine material. Based on the overprinting relationships of landforms on these uplands, we suggest that these ice caps dispersed lobes across the surrounding lowlands after regional-scale ice retreat and the drainage of proglacial lakes (Fig. 6). Although no chronology is available to constrain the timing of their operation, based upon their location relative to regional deglacial isochrones, ice from a dispersal centre on the Swan Hills could not have flowed across ice-free lowlands until after ~13.5 ka BP; similarly, ice on the Birch and Caribou Mountains and Cameron and Clear Hills, could not have dispersed radially until after ~13 ka BP. Based on this evidence, these upland ice caps may have formed as a localized response to YD cooling.

## 6 Discussion

### 6.1 Timing of ice sheet dynamic changes

Combining our ice dynamic reconstruction with existing chronologies, we reconstruct the behaviour of the SWLIS during the time period encompassing pre-LGM to the early Holocene. While switches in ice sheet dynamics and changes in ice-marginal behaviour have previously been recognized for the SWLIS (i.e., Ó Cofaigh et al., 2010; Ross et al., 2009; Margold et al., 2018; Evans et al., 2020), the duration over which these dynamic changes occurred and how they relate to climate forcing is less clear.

Integrating our flowset and ice-marginal geomorphic mapping with new deglacial chronologies, our reconstruction depicts three macroscale reorganizations of the ice drainage network followed by rapid ice sheet retreat and widespread stagnation that itself was punctuated by localized surging. These changes occurred over a ~2500 yr interval, synchronous with abrupt BA warming (14.7-12.9 ka BP). We, therefore, suggest that ice stream networks documented in the geomorphic record within the SWLIS evolved rapidly and were replaced by subsequent ice flows twice during this very short time period (see also Ross et al., 2009; Ó

Cofaigh et al., 2010). During the LGM, the SWLIS drained via three large north-to-south oriented ice streams that were unconstrained by the regional topography. South-eastward flowing, topographically constrained ice streams later replaced these unconstrained ice streams during a macroscale switch in ice sheet dynamics. We suggest this switch from unconstrained to topographically constrained ice flows occurred contemporaneously with rapid ice sheet thinning at 14.9-14.3 ka BP (Stoker et al., 2022). South-eastward ice streaming would only have been sustained for a short period while the Plains Ice Divide remained active to support flow from the northwest to the southeast (Ross et al., 2009; Ó Cofaigh et al., 2010; Margold et al., 2015b, 2018). Following the formation of an 'ice-free corridor' between the CIS and LIS at 14.9 ka BP (Margold et al., 2019), south-eastward ice flows would have ceased (Ross et al., 2009; Ó Cofaigh et al., 2010; Margold et al., 2015b, 2018). In the central portions of the SWLIS, we observe widespread ice sheet downwasting and effective collapse associated with the shutdown of south-eastward flows and the isolation of this region. Within the northern portions of the SWLIS, south-eastward ice stream corridors were replaced by south-westward oriented ice streams (Ross et al., 2009; Ó Cofaigh et al., 2010; Margold et al., 2015b, 2018) that fed the rapidly retreating ice margin between 14.5-13 ka BP (Norris et al., 2022). As the ice margin reaches the Canadian Shield boundary at ~13 ka BP, we observe a ~1500 yr ice marginal standstill of the SWLIS. This standstill occurred contemporaneously with YD (12.9-11.7 ka BP) cooling (Norris et al., 2022).

**6.2 Drivers of ice sheet dynamics**

Comparing our reconstruction with climatic driving forces, macroscale reorganization and deglaciation of the SWLIS occurred synchronously with climatic warming during the BA interval. It is likely that ice sheet thinning during this interval would have exerted a major control on the SWLIS dynamics (Ross et al., 2009; Ó Cofaigh et al., 2010; Margold et al., 2015b, 2018). First, our reconstruction documents an initial switch in ice flow orientation from large topographically unconstrained ice streams that flowed north to south (CAIS, HPIS, Maskwa IS) to southeastward ice streaming (IS2 complex), the orientation of which was controlled by the regional subglacial topography and fed by the Plains Ice Divide (Ross et al., 2009; Ó Cofaigh et al., 2010; Margold et al., 2015b, 2018). In agreement with previous work, we suggest that ice sheet thinning likely impeded north-to-south ice flow due to there being a lack of adequate ice thickness to sustain topographically unconstrained flows (Ross et al., 2009; Ó Cofaigh et al., 2010; Margold et al., 2015b, 2018). A second macroscale ice sheet reorganization within the northern portions of the SWLIS was characterized by a switch to southwestward ice streaming (Hay, Peace, and Athabasca river IS) (see also Ross et al., 2009; Margold et al., 2015b). We suggest this switch was driven by further ice sheet thinning (Ross et al., 2009), the shutdown of the Plains Ice Divide and the separation of the LIS and CIS (Margold et al., 2018). As the two ice sheets thinned and separated, there would have been inadequate ice input from the northwest to feed ice streaming to the southeast, and thus these ice flowpaths would have been replaced by ice streams that travelled through the Hay, Peace, and Athabasca river lowlands fed solely by the Keewatin Ice Dome (Ross et al., 2009; Margold et al., 2015b, 2018).

In both cases, ice stream dynamics were likely driven by ice sheet thinning and changing ice divide configuration. However, these changes alone would not have caused these dramatic switches in ice stream orientation. Instead, these macroscale changes in ice sheet dynamics observed in the SWLIS are likely moderated by the region's subglacial topography (i.e., preglacial valley network). As the ice sheet thinned, the subglacial topography in the region ultimately controlled the direction of ice stream switching (Ross et al., 2009; Ó Cofaigh et al., 2010). As such, this control, internal to the ice sheet system, influenced the wider response of the SWLIS to BA warming.

Our reconstruction also depicts a distinct change in the dynamics of the SWLIS during the YD. During this period of climatic cooling, we observe a slowdown in the ice sheet retreat rate and a widespread standstill of the SWLIS marked by the formation of the regionally extensive Cree Lake Moraine complex (Norris et al., 2022). At the same time, we record a switch from topographically constrained ice streaming in the Hay, Peace, Athabasca, and Churchill river lowlands to a regionally extensive and unconstrained sheet flow that fed the ice margin at its Cree Lake Moraine position (Margold et al., 2018). While it is highly likely that these dynamics were influenced by the external climate forcing of the YD, the regions' underlying lithology and topography likely moderated the ice sheets' response to the external forcing (Margold et al., 2018). Interestingly, these changes in ice flow behaviour and regional ice margin standstill occur at the transition from the soft (deformable) sediment-covered Western Canadian Sedimentary Basin to the predominantly hard (rigid) bed of the crystalline Canadian Shield (Margold et al., 2018). While we acknowledge that in some areas, the Canadian Shield is mantled by sediment, the general change in the underlying bed composition (from predominantly soft bedded to predominately hard bedded) likely resulted in an increase in bed strength and frictional resistance at the ice-bed interface, resulting in a decrease in ice velocity and, thus, potentially a slowing of the rate of ice sheet retreat perhaps even independent of external climate forcing (Bradwell et al., 2019). This pattern is analogous to changes observed in the northern portions of the British Irish Ice Sheet (Bradwell et al., 2019; Clark et al., 2022a), where bedrock lithology played a similar role in changing ice sheet dynamics. Furthermore, it is plausible that a shift from retreat down a reverse bed slope over the Western Canadian Sedimentary Basin, with large time-transgressive ice-marginal glacial lakes (see Fig 6), to retreat across a normal bed slope, with less extensive impounded lakes on the Crystalline Shield also decreased retreat rates (e.g., Stokes and Clark, 2004; Evans et al., 2012; Utting and Atkinson, 2019). This change in the density of ice-marginal lakes may have resulted in changes in water availability at the ice-bed interface, which could have indirectly influenced ice dynamics (Carrivick and Tweed, 2013). The simultaneous change in lithology, bed slope, frequency of ice-marginal lakes and climate experienced by the SWLIS is in contrast to many other portions of the LIS at this time and may therefore help explain why this sector of the LIS has such a marked response to YD cooling, especially in comparison to other portions of the western Laurentide margin (e.g., Gauthier et al., 2022; Reyes et al., 2022).

The dynamics we highlight above attest to the importance of considering glaciological, lithological and topographic factors in addition to external climate forcing when assessing ice sheet dynamic evolution over time. The interplay between internal and external forcings of a terrestrial terminating ice margin has been highlighted for other ice sheets during the last glacial period (e.g., Landvik et al., 2014; Davies et al., 2019; see section 6.4). However, the changes in ice stream orientation, ice margin stagnation and localized surging highlighted in our reconstruction have not yet been well replicated by the existing numerical ice sheet reconstructions for the SWLIS (e.g., Tarasov et al., 2012). Quantifying and incorporating the effects of internal as well as external controls into numerical model reconstructions of the SWLIS should be a key area for future work to address.

**6.3 The geomorphic imprint of abrupt climate change**

In addition to the observed changes in ice flow direction, our geomorphic reconstruction also provides insight into the pattern and mechanism of ice recession. We observe a geomorphic imprint that signifies a period of active recession, widespread ice sheet stagnation punctuated by localized surging and then a return to widespread active recession (see Fig. 5 and 7). The spatial variability in landforms diagnostic of these changes is best described as three zones. A southern 'Outer Deglacial Zone' encompasses the region between the LGM ice margin position and its retreat as far north as Red Deer (~52°N), an 'Intermediate Deglacial Zone' encompasses the central regions of Alberta and Saskatchewan spanning from Red Deer to Cold Lake (~55°N) and an 'Inner Deglacial Zone' that spans the most northern portion of the SWLIS and includes the region from Cold Lake to the

Cree Lake Moraine complex. The juxtaposition of these three zones is likely the result of macroscale ice flow reorganizations that occurred during ice sheet thinning and deglaciation.


### 6.3.1 Outer Deglacial Zone

At the southern limits of the SWLIS, recessional push moraines are common across the landscape, most significantly the Lethbridge Moraine and the recessional moraines that reach the northern side of Cypress Hills (Westgate, 1968; Stalker, 1977; Kulig, 1996; Evans et al., 2008, 2012, 2014; Fig. 5 and 6a). Multiple studies have invoked this landform signal to represent active recession of

ice feeding the HPIS, CAIS, and Maskwa IS until they had retreated to the region north of Red Deer (Evans et al., 2008, 2014; Fig. 4d). Outside of these ice stream tracks, active recession is marked by arcuate zones of 'aligned hummocks, ponds and ice-walled lake plains' (controlled moraine) diagnostic of warm and polythermal ice-marginal conditions (Evans et al., 2014; Fig. 3, 2020). This region also contains very limited glaciolacustrine sediments and landforms, which have previously been interpreted to be indicative of small, very short-lived proglacial lakes (Utting and Atkinson, 2019). We suggest this is the result of rapid active-ice

sheet recession, which inhibited long-lived, widespread lake formation.

### 6.3.2 Intermediate Deglacial zone

The geomorphic record in central Alberta and Saskatchewan (i.e., north of Red Deer) contains evidence of widespread stagnation punctuated by localized surging (Fig. 5 and 7b). Within the central portions ('ice stream trunk zones') of the CAIS and Maskwa

IS and the south-westward IS2 complex that crosses the region, ice sheet stagnation is demarcated by regions of well-preserved lineations (drumlinoid ridges and MSGL), which are overprinted by crevasse fill ridges (Ó Cofaigh et al., 2010). Bordering the flowpaths of these ice streams, large regions of inter-ice stream moraine containing hummocky terrain dominate the landscape (Evans et al., 2008, 2020). In some locations, glaciotectonic landforms, small localized moraines, and streamlined zones that overprint the palaeo-ice stream imprint are interpreted as the product of local late-stage ice surging (Bretz, 1943; Evans, 2000;

Evans et al., 2008; Ó Cofaigh et al., 2010; Evans et al., 2014, 2016; Atkinson et al., 2016; Evans et al., 2020). We suggest that local scale surging that punctuated widespread stagnation was most likely conditioned by the increased availability of subglacial meltwater and the development of local proglacial lakes around areas that contained downwasting ice (see also Evans et al., 2008, 2020).

We propose that the intermediate deglacial zone results from an 'ice sheet collapse' (ice sheet sector isolation, stagnation, and downwasting) caused by ice stream switching and subsequent shutdown (Fig. 7b). This geomorphic signature was previously highlighted by Ross et al. (2009) and Ó Cofaigh et al. (2010) and we support the notion that this macroscale ice flow reorganization would have caused stagnation in higher topographic regions. These regions would have become separated from the ice sheet accumulation zone as ice flows shifted and began to flow in a southeast direction in topographically confined low-elevation areas

(Ross et al., 2009; Ó Cofaigh et al., 2010). Following rapid ice sheet thinning, these southeast flowpaths would also have ceased, and ice would have downwasted, resulting in ice stagnation, including within the former ice stream flowpaths (Ross et al., 2009; Ó Cofaigh et al., 2010). In some locations, small local surge lobes would have occurred, overprinting the underlying geomorphology (Fig. 7c).

Although widespread ice sheet stagnation and downwasting (initial ice sheet collapse) would have been rapid, the landscape may have remained 'ice-cored' for several thousand years (Fig. 7c and d). In such regions, the thinning of the ice sheet would have allowed permafrost to aggrade beneath it and persist in the ice-cored terrain. Therefore, many regions of ice-cored hummocky

terrain southwest of the Cree Lake Moraine, while formed during rapid ice downwasting, would have continued to be modified by the thawing of buried glacier ice into the Early Holocene. For example, in central Alberta in the Beaver Hills/ Cooking Lake region (20 km east of Edmonton), Emerson (1983) suggests glacial ice melt and permafrost thaw continued into the early Holocene and persisted until 9 ka BP based on radiocarbon ages from postglacial freshwater molluscs.

### 6.3.3 Inner Deglacial zone

Within the northern portions of Alberta and Saskatchewan, suites of moraines, eskers, and meltwater channels are visible within the Churchill, Athabasca, and Peace river lowlands (Fig. 5 and 7c). These landforms document active retreat of the Hay, Peace, and Athabasca River ice streams and IS 4/5 (Fig. 3b) (Margold et al., 2018, 2015b). In these regions, hummocky terrain is scarce and mostly restricted to upland regions (e.g., the Caribou and Birch Mountains and the Buffalo Head Hills) (see Paulen and McClenaghan, 2015; Atkinson et al., 2018). Active ice retreat can be traced northeast, following the long axis of each valley as far north as the Canadian Shield boundary. The ice sheet retreat slowed to a standstill at this location to form the regionally extensive Cree Lake Moraine complex.

The marked change in deglacial behaviour observed across the SWLIS demonstrates the instability of a terrestrially terminating ice margin under rapid climatic warming. The behaviour of the 'Intermediate Deglacial Zone' was the result of macroscale ice stream reorganization that isolated a large portion of the SWLIS and forced an effective collapse in this region. This ice stream response was triggered by the rapid thinning of the CIS-LIS convergence zone, collapse of the CIS-LIS saddle, and final shutdown of the Plains Ice Divide (Margold et al., 2018) and thus highlights how the ice sheet, through changes in the ice flow regime, can be highly sensitive to rapid climatic change. Although the response of the SWLIS to climatic change is complex, we suggest that its geomorphic imprint identified here is not unique and propose that the combination of regions of rapid ice marginal retreat and ice sheet disintegration and downwasting may be a valuable indicator of terrestrial ice sheet collapse, especially in soft-bedded regions.

### 6.4 Comparison to other ice sheets dynamics

The documented changes within the SWLIS provide valuable insight into the dynamics of ice streams in a region associated with a terrestrially terminating ice sheet margin during a period of rapid climate change. Despite its much larger scale, the behaviour of the SWLIS exhibits similarities to the deglacial changes observed within the interior portions of the last British Irish Ice Sheet (Greenwood and Clark, 2009; Davies et al., 2019; Clark et al., 2022a) and the (Late Weichselian) Svalbard-Barents Sea Ice Sheet (Landvik et al., 2014). These ice sheets exhibited relatively rapid, major changes in ice sheet and ice stream dynamics during a single glaciation. In all three cases, at the local last glacial maximum, the ice sheet dynamics were predominantly driven by changes in external climatic forcings, such as changes in global temperatures and regional precipitation patterns, and were less sensitive to internal controls, such as topography or lithological variations (Landvik et al., 2014; Davies et al., 2019). However, as noted by Davies et al. (2019), associated with a decrease in an ice sheet's thickness, lithological, and topographic controls start to play a more dominant role, modifying the macroscale changes in ice flow dynamics that were initially a response to the external climatic forcing.  Based on our reconstruction of the SWLIS and the examples provided above, we suggest that while external climatic forcing is often implicated as a driver of change in ice sheet and ice stream dynamics at terrestrially terminating ice sheet margins, such changes may not result implicitly from an external control alone. Instead, these ice sheet dynamics result from a combination of external climatic forcing, as well as internal glaciological, lithological, and topographic controls (Greenwood and Clark, 2009;

Ross et al., 2009; Ó Cofaigh et al., 2010; Landvik et al., 2014; Margold et al., 2015b; Davies et al., 2019) and thus both should be considered.

## 7 Conclusion

Using glacial geomorphological inversion methods (flowsets) in combination with a recently compiled regional chronological synthesis, we reconstruct the ice flow and ice-marginal retreat dynamics of the SWLIS into five snapshots of ice sheet behaviour that encompass the time period from pre-LGM to the early Holocene. Our reconstruction shows ice stream behaviour comprising three macroscale reorganizations in the orientation and dynamics of ice streaming followed by regional deglaciation associated with a rapid increase in northern hemisphere temperatures during the BA. Early, faintly preserved, westward flows are documented

by a small number of flowsets and likely demarcate an early stage of ice streaming before the LGM. At the LGM our flowsets document the southward flow of the CAIS, HPIS, and Maskwa Ice Stream. Linked to ice sheet thinning at ~15 ka, a macroscale reorganization in ice stream flow occurred, resulting in a series of southeastward flows. A second macroscale reorganization in ice flow is then observed at ~14 ka, where southwestern ice flow is restricted to the Hay, Peace, Athabasca, and Churchill river lowlands. During the YD, rates of ice sheet retreat then slowed. At this time, the ice sheet margin was located north of the Canadian

Shield boundary, and ice flow continued to be sourced from the northeast. Resulting from these phases of ice sheet dynamics, we observe a three-part zonation of ice-marginal geomorphology within the SWLIS, that demarcates active ice margin recession followed by punctuated ice sheet stagnation and downwasting (terrestrial ice sheet collapse). We attribute these macroscale changes in ice flow geometry and the associated deglacial behaviour to external climatic controls during the BA and YD. We also highlight the concordant role of internal (glaciological, lithological, and topographic) controls in influencing how an ice sheet

responds to external climate controls.

**Data availability.** The data referred to in this paper is provided within the tables and figures in the main text and in the supplementary materials related to this article are available in the Supplement and at https://figshare.com/s/4270b557926dd0806938. LiDAR DEMs were provided by the Government of Alberta (2017). LiDAR

Data Archives. These data were provided under license and with project-specific data-sharing agreements by the Archaeological Survey of Alberta, Culture, Multiculturalism, and the Status of Women. Edmonton, Alberta via geodiscoveralberta@gov.ab.ca through https://geodiscover.alberta.ca/ geoportal/#searchPanel. SRTM DEMs were accessed via the USGS EROS Archive (EROS Centre, 2018) (https://www.usgs.gov/centers/eros/science/usgs-eros-archive-digital-elevation-shuttle-radar-topography-mission-srtm-non, last accessed Aug 15th 2023).

**Supplement.** The supplement related to this article is available online at: https://figshare.com/s/4270b557926dd0806938

**Author contributions.** This project was conceptualized by DGF and SN with input from MM. Geomorphic mapping and flowset inventories were developed by SLN, NA and DJAE. The data visualization and figure creation were completed by SLN with input from all authors The manuscript was written by SLN with input from all authors.

**Competing interests.** The contact author has declared that none of the authors has any competing interests.

**Acknowledgements.** We are grateful for comments and discussion from B. Menounos, J. England and A. Reyes on an early version of this manuscript. Internal reviews from M. Grobe and D. Utting at the Alberta Geological Survey helped clarify the manuscript. We thank reviewers Samuel Kelley and Isabelle McMartin for their helpful reviews that improved the quality of this paper.

**Financial support.** This research was funded by the Natural Sciences and Engineering Research Council and the Canada Research Chairs Program awarded to DGF, the Czech Science Foundation grant no. 19-21216Y awarded to MM, and grants from the University of Alberta Northern Research Awards to SLN.

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

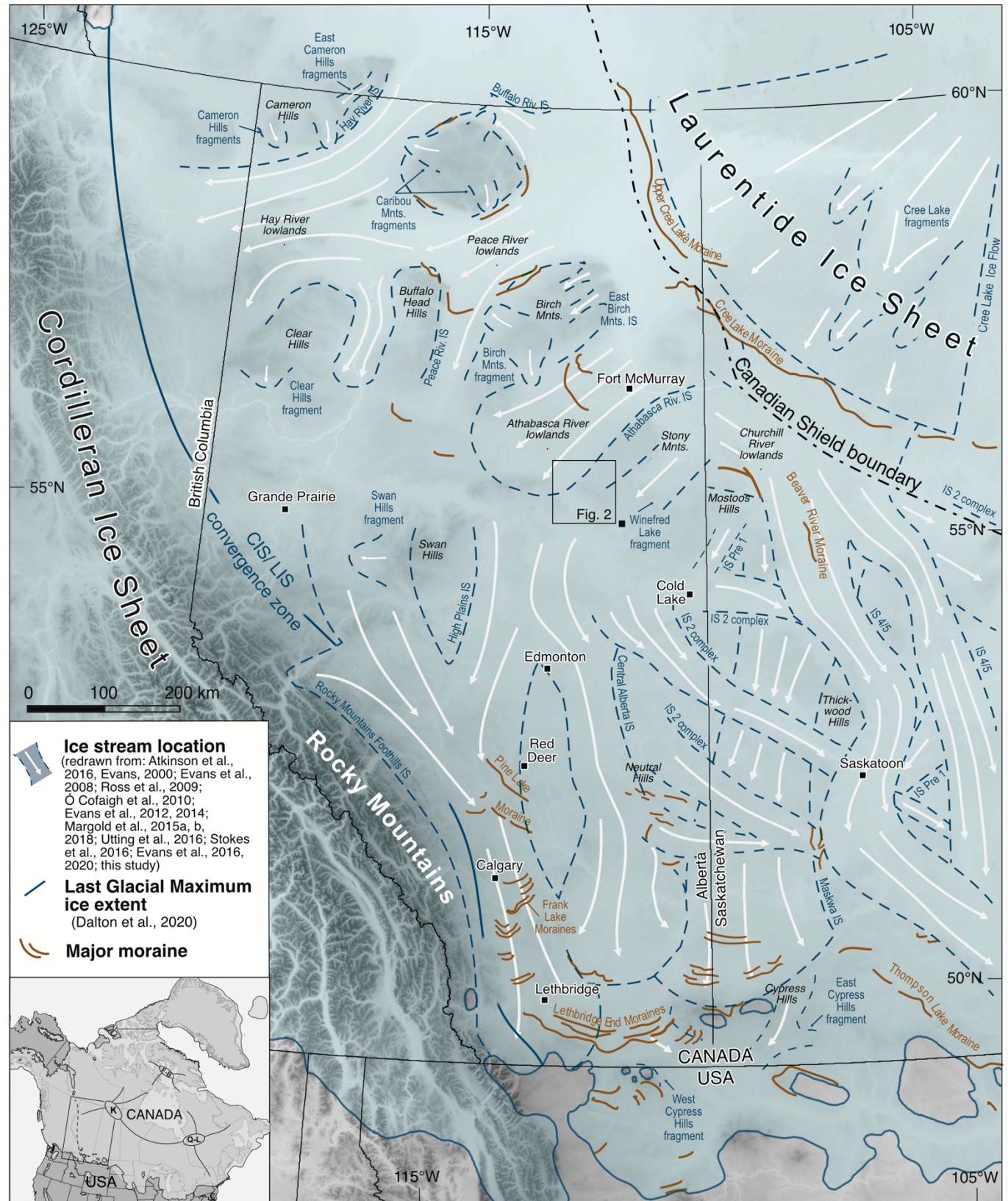

**Figure 1: Regional ice-flow imprint of the southwestern sector of the Laurentide Ice Sheet. Digital elevation model (DEM) from the Shuttle Radar Topographic Mission (SRTM) provided by the USGS EROS Archive (EROS Centre, 2018). Last Glacial Maximum ice extent from Dalton et al. (2020).**

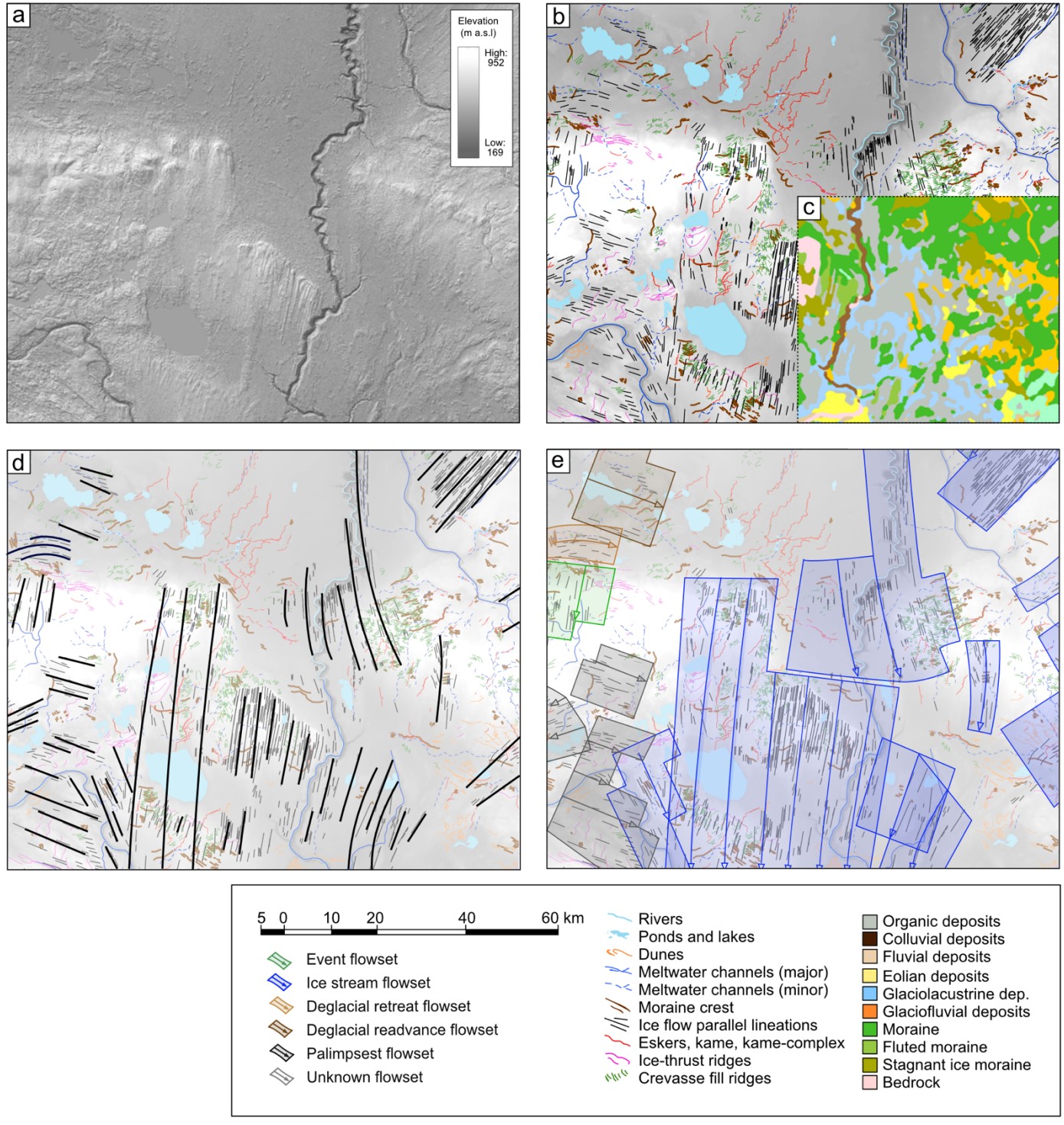

**Figure 2: Classification of glacial geomorphology into flowsets. (a) Imagery displaying the onset zone of the Central Alberta Ice Stream. (b) Area of corresponding geomorphological mapping (Atkinson et al., 2018) and, (c) surficial geology (Fenton et al., 2013) (d). Simplification of the geomorphic imprint into flowlines, and (e) flowsets. Flowsets were categorized as either 'ice stream', 'deglacial retreat', 'deglacial readvance' 'event' 'palimpsest' or 'unknown' after Kleman (2006). See Figure 1 for the location of Figure 2 panel. Digital elevation model (DEM) from the Shuttle Radar Topographic Mission (SRTM) provided by the USGS EROS Archive (EROS Centre, 2018).**

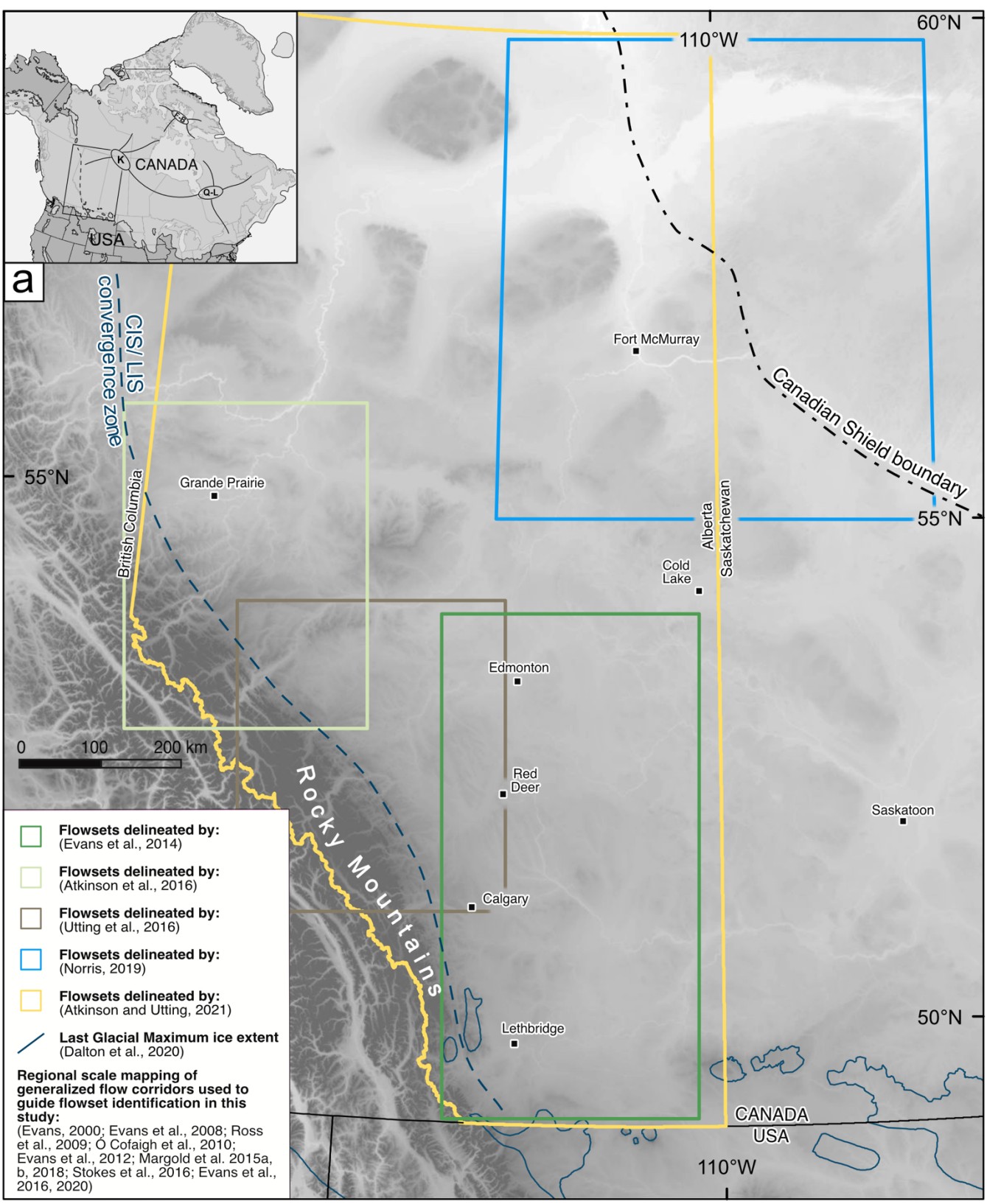

a

CIS/LIS convergence zone

British Columbia

Canadian Shield boundary

60°N
110°W

Fort McMurray

55°N

Grande Prairie

Alberta
Saskatchewan

55°N

Cold Lake

0    100    200 km

Edmonton

Red Deer

Saskatoon

Rocky Mountains

Calgary

50°N

Lethbridge

CANADA
USA

110°W

**Flowsets delineated by:**
(Evans et al., 2014)

**Flowsets delineated by:**
(Atkinson et al., 2016)

**Flowsets delineated by:**
(Utting et al., 2016)

**Flowsets delineated by:**
(Norris, 2019)

**Flowsets delineated by:**
(Atkinson and Utting, 2021)

**Last Glacial Maximum ice extent**
(Dalton et al., 2020)

**Regional scale mapping of generalized flow corridors used to guide flowset identification in this study:**
(Evans, 2000; Evans et al., 2008; Ross et al., 2009; Ó Cofaigh et al., 2010; Evans et al., 2012; Margold et al. 2015a, b, 2018; Stokes et al., 2016; Evans et al., 2016, 2020)

CANADA

USA

K

F-B

Q-L

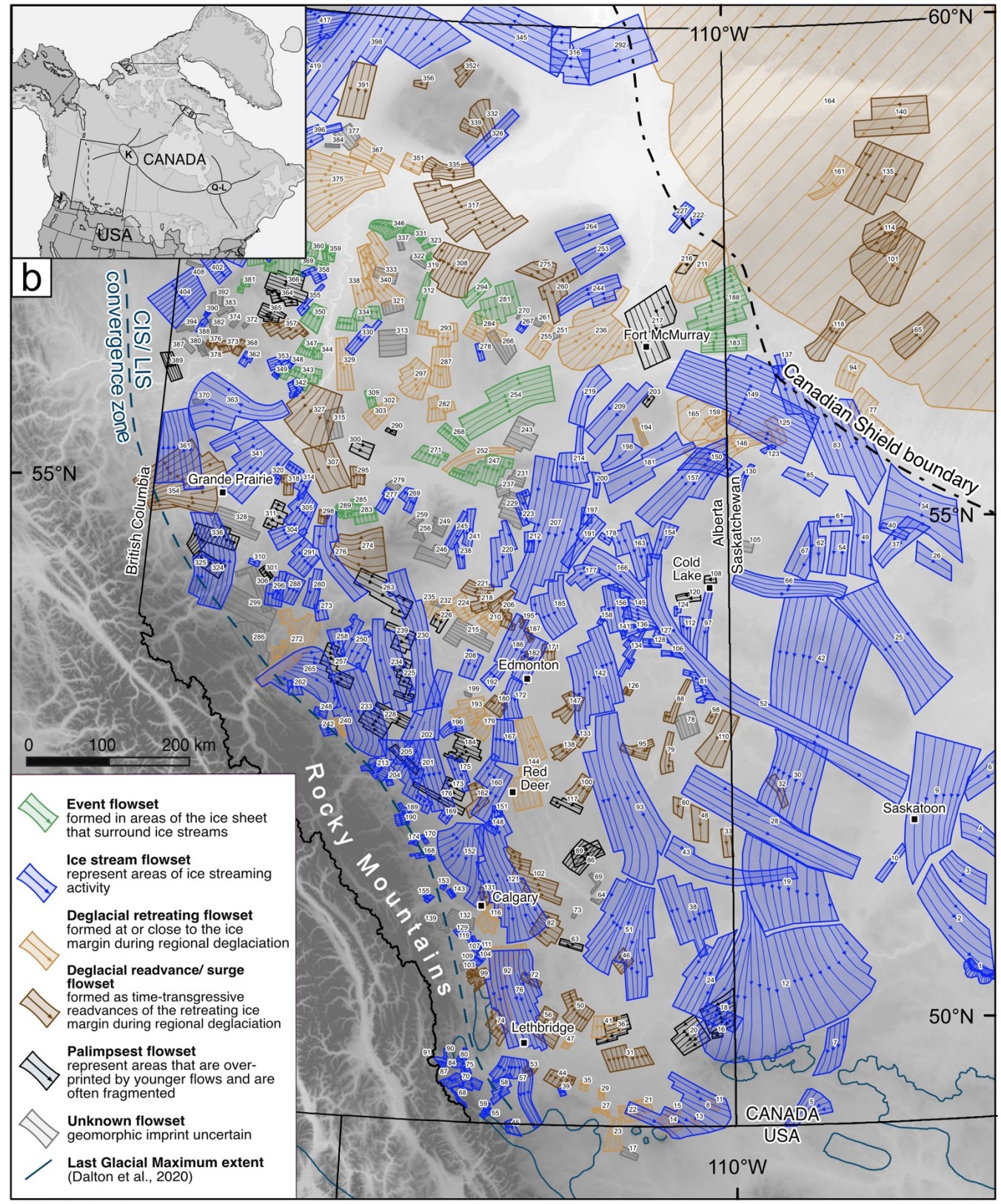


**Figure 3: (a) Location of previous flowset mapping. (b) Flowset map of the southwestern Laurentide Ice Sheet (available as a supplementary .kml file). Digital elevation model (DEM) from the Shuttle Radar Topographic Mission (SRTM) provided by the USGS EROS Archive (EROS Centre, 2018). Last Glacial Maximum ice extent from Dalton et al. (2020).**

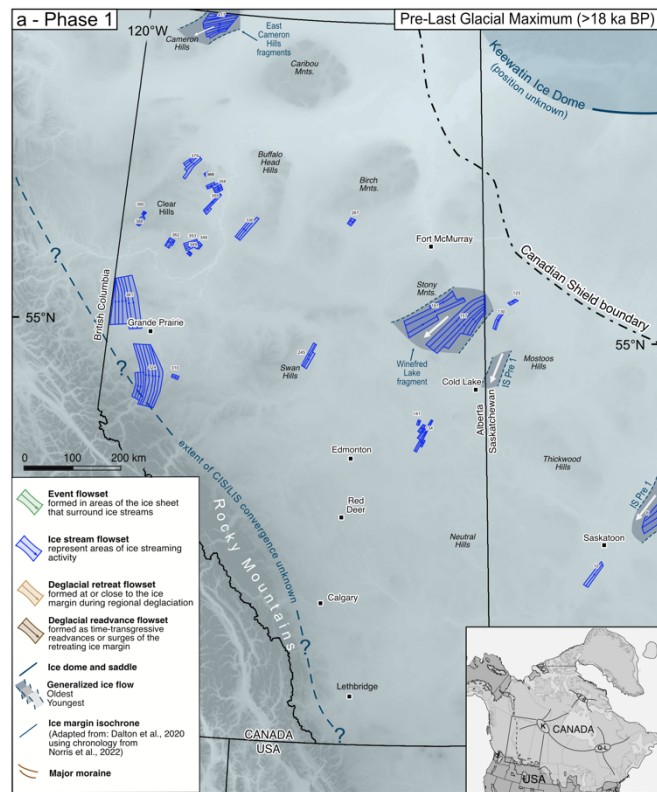

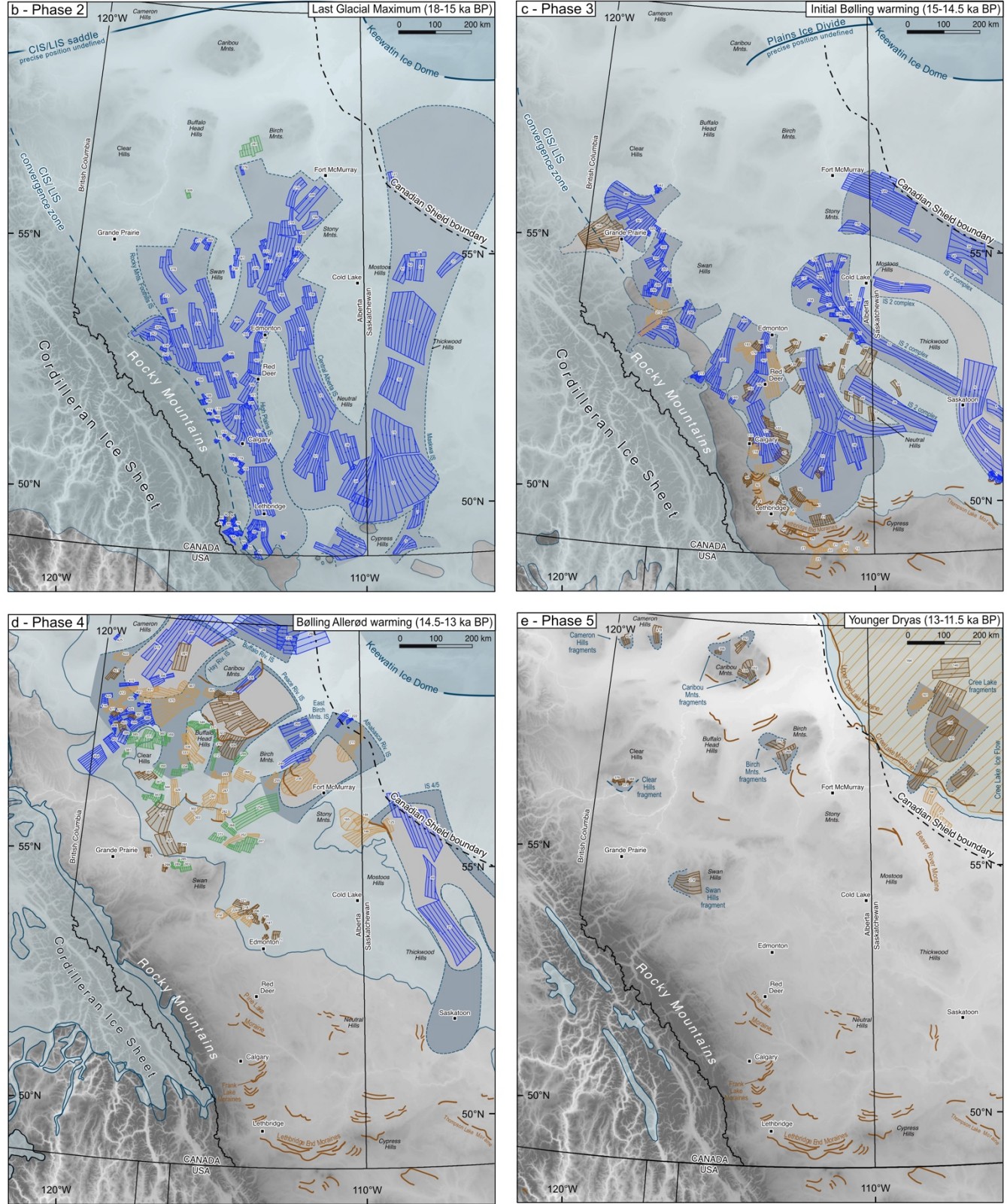


**Figure *4*: Deglacial ice flow dynamics of the southwestern Laurentide Ice Sheet. This reconstruction presents a comprehensive synthesis, collating pre-existing ice stream mapping with ice streams presented in this study. Timing of ice retreat and associated ice flow reorganizations are constrained using the deglacial chronology of Norris et al. (2022). Digital**

**elevation model (DEM) from the Shuttle Radar Topographic Mission (SRTM) provided by the USGS EROS Archive**

**(EROS Centre, 2018). Last Glacial Maximum ice extent from Dalton et al. (2020).**

*Note Figure 4 is a five-panel figure.*

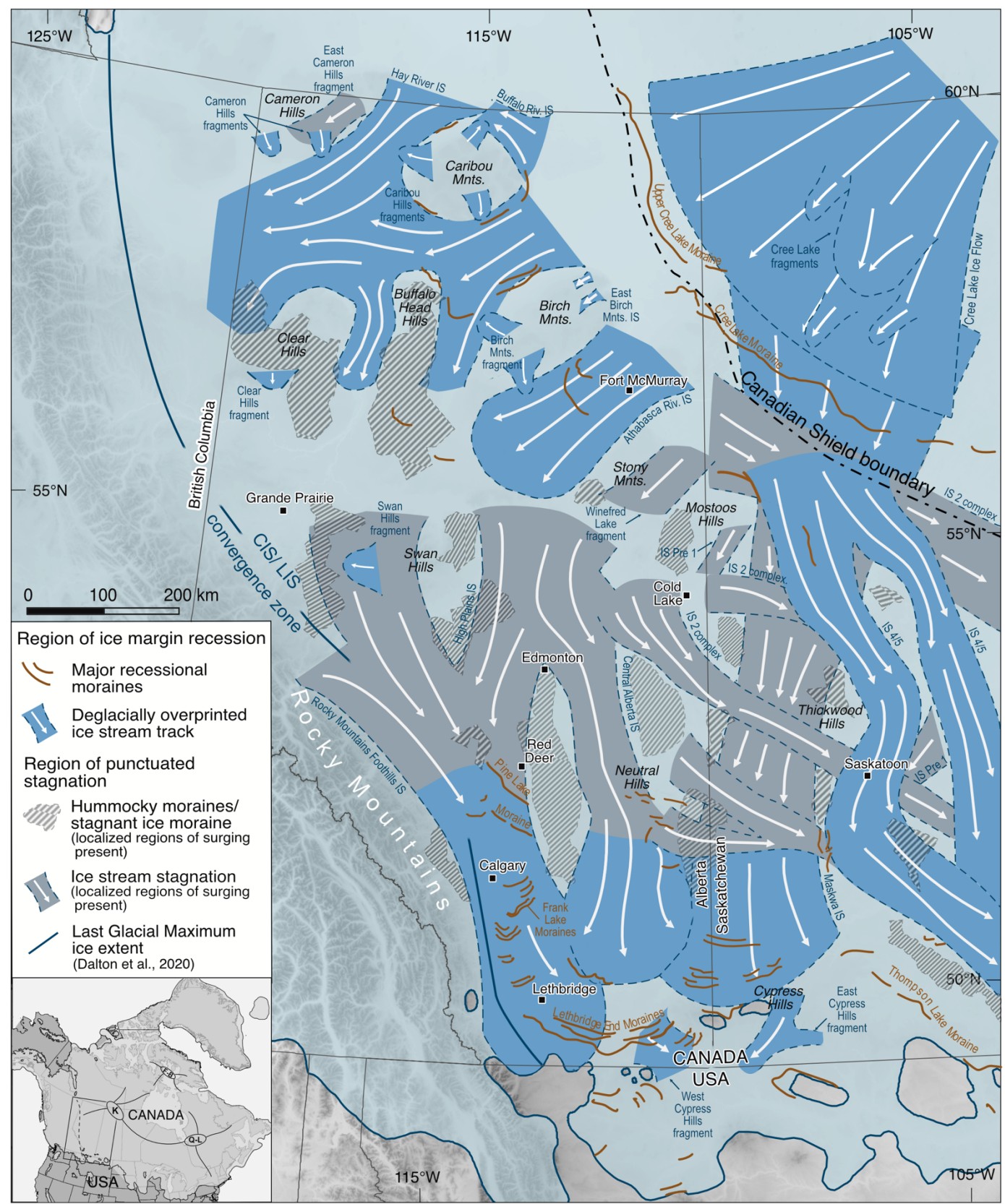

**Figure 5: Regional geomorphic imprint of the distribution of ice sheet active retreat and punctuated stagnation across the southwestern Laurentide Ice Sheet. Digital elevation model (DEM) from the Shuttle Radar Topographic Mission (SRTM) provided by the USGS EROS Archive (EROS Centre, 2018). Last Glacial Maximum ice extent from Dalton et al. (2020).**


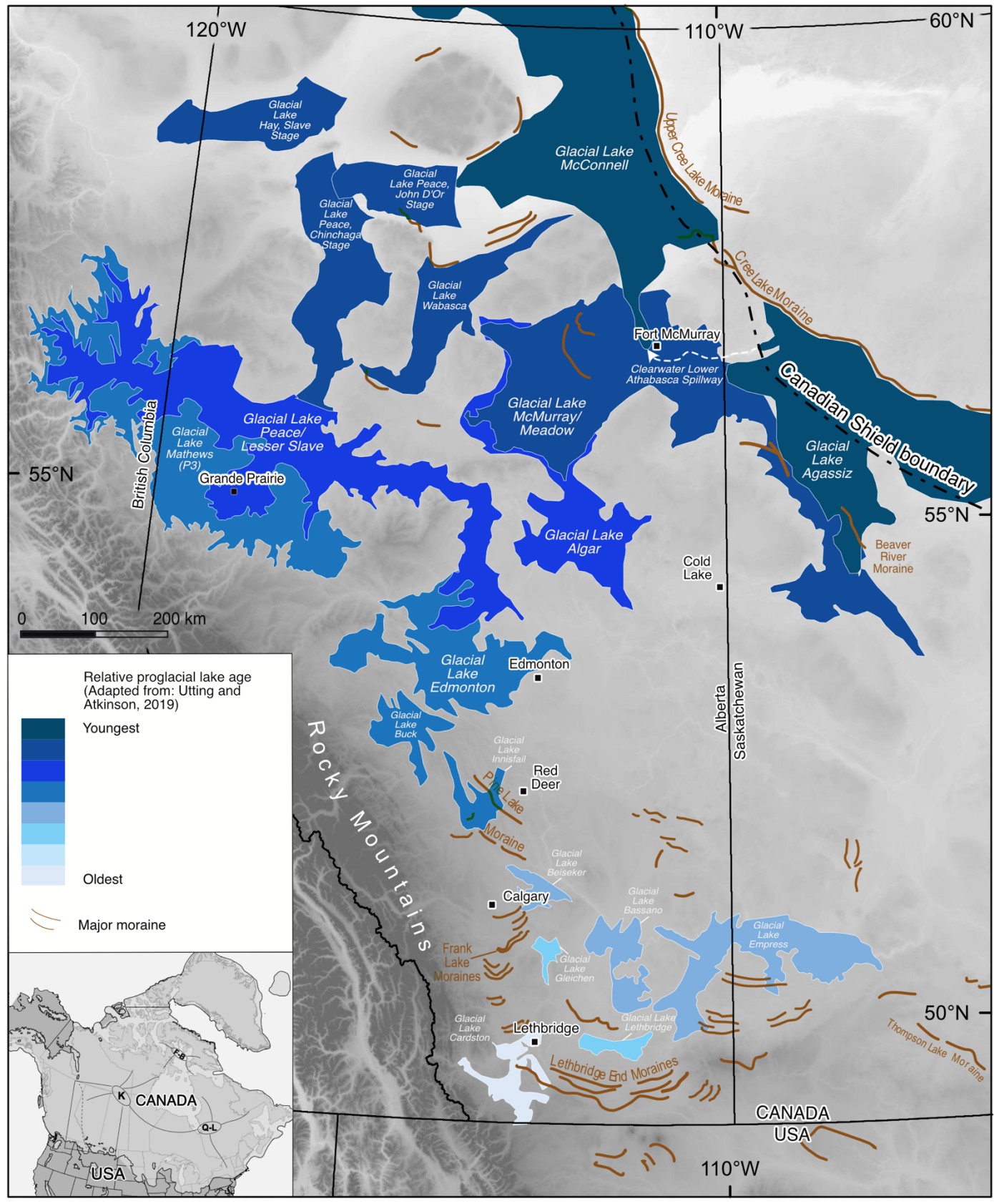

**Figure 6: Distribution of reconstructed proglacial lakes associated with the retreat of the southwestern Laurentide Ice Sheet, classified by relative age adapted from Utting and Atkinson (2019). Digital elevation model (DEM) from the Shuttle**

Radar Topographic Mission (SRTM) provided by the USGS EROS Archive (EROS Centre, 2018). Last Glacial Maximum
ice extent from Dalton et al. (2020).

### a  INITIAL ICE MARGIN RETREAT

**A- Outer Zone**
Active ice margin recession

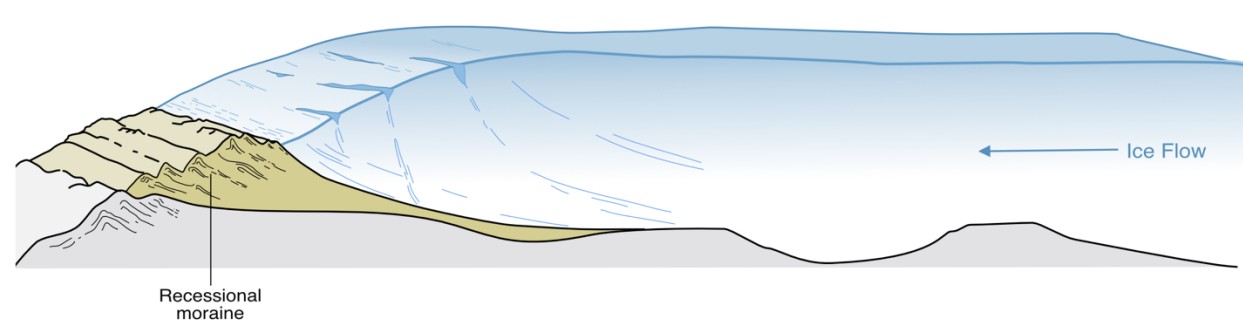

Recessional moraine

Ice Flow

### b  DEGLACIAL ICE FLOW SWITCHING & ICE MARGIN ISOLATION

**A- Outer Zone**
Active ice margin recession

**B- Intermediate Zone**
Ice stagnation and downwasting,

Ice Flow

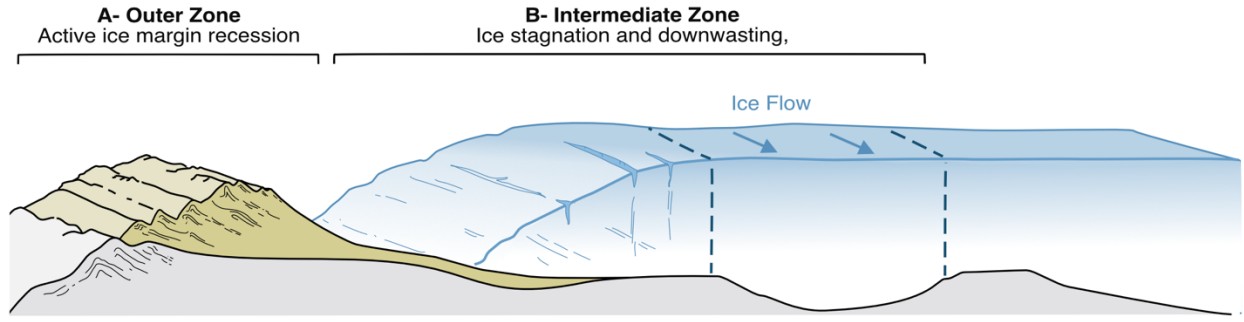

### c  PUNCTUATED ICE STAGNATION AND DOWNWASTING

**A- Outer Zone**
Active ice margin recession

**B- Intermediate Zone**
Ice stagnation and downwasting

Ice Flow

Hummocky moraine

Localised glacial surging

Localised minor recessional moraines

Dead-ice

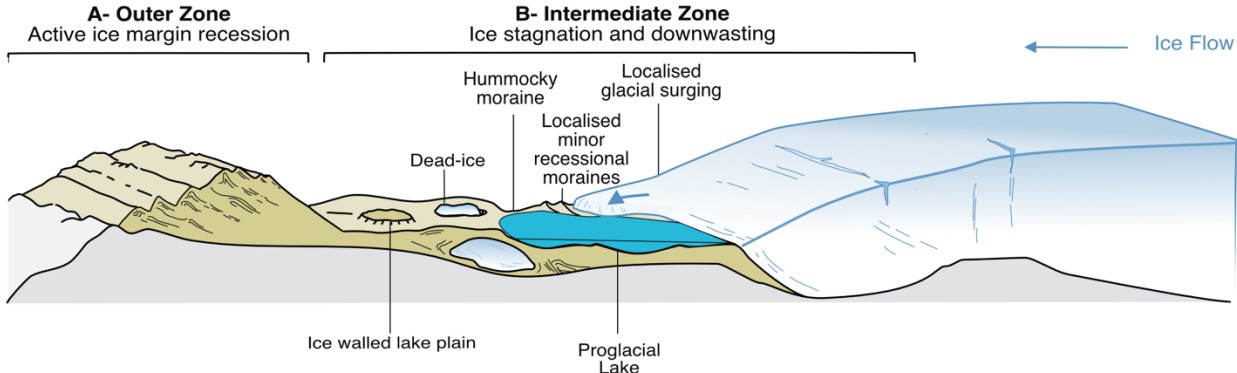

Ice walled lake plain

Proglacial Lake

### d  LATE STAGE ACTIVE MARGIN RETREAT

**A- Outer Zone**
Active ice margin recession

**B- Intermediate Zone**
Localised ice stagnation and downwasting

**C- Inner Zone**
Active ice margin recession

Esker

Crevasse-squeeze ridge

Ice Flow

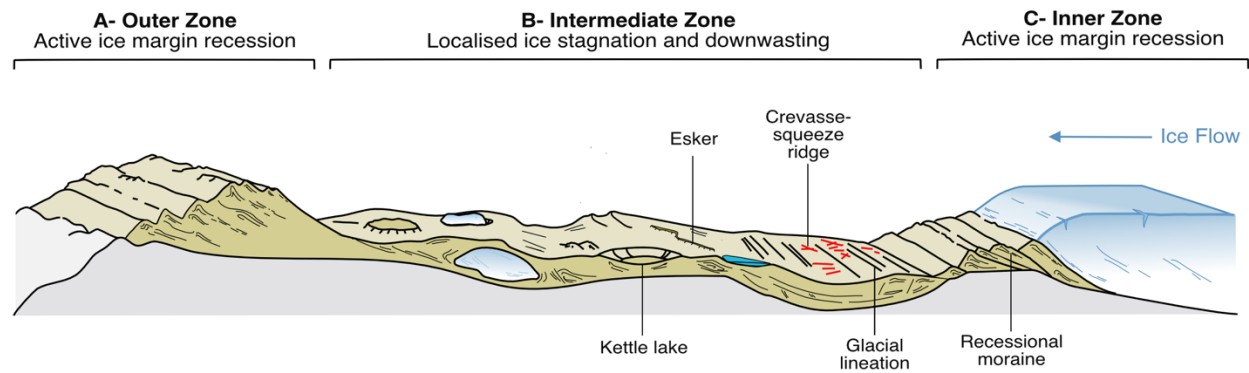

Kettle lake

Glacial lineation

Recessional moraine

**Figure 7: Schematic model of the geomorphic imprint associated with the terrestrial ice sheet margin collapse of the southwestern Laurentide Ice Sheet. (a) Shows initial ice margin retreat in the outer deglacial zone. Large ice streams fed the margin and are orientated perpendicular to the direction of marginal retreat. (b) Shows deglacial ice flow switching and reorganization resulting in the isolation of large regions of the ice sheet in the intermediate deglacial zone. (c) Ice sheet thinning, and rapid retreat triggered macroscale ice stream reorganization, resulting in the isolation of large regions of the ice sheet and ice sheet stagnation in the intermediate deglacial zone. In some locations, stagnation terrain is crosscut by the signature of local small surge events. (d) Shows late-stage active margin retreat of the inner deglacial zone.**

**Table 1: Diagnostic criteria for identifying flowset types (after Kleman et al., 1997; Stokes and Clark, 1999; Kleman et al., 2006; Greenwood and Clark, 2009).**

| Flowset types | Characteristics |
|---|---|
| Ice stream | Represent corridors within an ice sheet that are flowing faster relative to the surrounding ice. Composed of groups of ice flow parallel lineations (drumlinoid ridges, mega-scale glacial lineations (MSGL), and crag-and-tail features) with abrupt lateral margins (in places marked by lateral shear moraines). |
| Event | Represent areas of ice flow surrounding corridors of fast flow (ice streams). Composed of groups of ice flow parallel lineations (drumlinoid ridges, mega-scale glacial lineations (MSGL), and crag-and-tail features). These flowsets are not associated with meltwater channels, but may contain eskers that overprint the flowset. These eskers are commonly misaligned with the flowset and result from a later stage of deglaciation, which are associated with different flowset orientation. |
| Deglacial retreat | Represent areas of the ice sheet at or close to the margin that underwent active (warm-based) retreat. Composed of groups of ice flow parallel lineations (drumlinoid ridges, mega-scale glacial lineations (MSGL), and crag-and-tail features) which align locally with eskers and meltwater channels. These flowsets are often associated with terminal, recessional moraines or glaciotectonic thrust ridges. |
| Deglacial readvance | Represent areas of the ice sheet at or close to the margin that underwent active (warm-based) retreat and were then locally overprinted during a minor readvance or change in ice flow direction during retreat. In some cases, these flowsets may represent switches between periods of rapid and slow ice flow driven by internal forcing mechanisms, in such cases we refer to them as exhibiting surge behaviour. Composed of groups of ice flow parallel lineations (drumlinoid ridges, mega-scale glacial lineations (MSGL), and crag-and-tail features) which align locally with eskers and meltwater channels. These flowsets are often associated with terminal, recessional moraines or glaciotectonic thrust ridges. |
| Palimpsest | Represent areas that are overprinted by younger ice flows with older flow traces exhibiting a similar orientation. These flowsets are often fragments. |
| Unknown | Represent areas where the age and significance of landforms are uncertain. These flowsets are often fragments or contain insufficient landforms to interpret their origin. |