# Peer review of "Dynamical response of the southwestern Laurentide Ice Sheet to rapid Bølling-Allerød warming"

_The Cryosphere, 2023_

## Author Comment (AC1)

We thank the editor and reviewers for their constructive comments on this manuscript. Below, we explain the changes we have made to the manuscript and answer in detail all the comments made by the reviewers. Comments made by the reviewers are in black, and our answers are in blue.

**Reviewer 1 comments:**
We thank Reviewer 1 for the constructive comments on the manuscript. Below we outline the changes made in reference to the reviewer's text comments.

This is a well-presented study that compiles and synthesizes a large amount of geomorphic data from the southwestern sector of the Laurentide Ice Sheet, placing their geomorphic reconstruction in the context of recent advancements in our understanding of the chronology of regional ice sheet recession. In all, the manuscript is well structured and written and pulls together a wealth of data that is of interest to the wider community. I would recommend this for publication and list a few minor suggestions for improvement below:

_Broader comments:_
The interpretation of a change in ice-marginal behaviour during BA to YD period connected to a change in underlying lithology is quite interesting but somewhat lacks explanation. The change in subglacial lithology will drive a change in friction at the ice sheet's bed, but also potentially yields a change in water availability at the bed-ice interface (moving from more porous sedimentary rocks to less porous crystalline rocks). The observation is worth noting, I feel the interpretation requires a little bit more discussion.

We agree with Reviewer 1's broader comment and have now added additional details regarding the role of meltwater and proglacial lakes in affecting ice velocity and retreat rates. We do not comment on the porosity of the rock types but feel it is more important to highlight the role of reserve topography and the formation of proglacial lakes because these features would have been instrumental in affecting ice sheet dynamics.

_"Our reconstruction also depicts a distinct change in the dynamics of the SWLIS during the YD. During this period of climatic cooling, we observe a slowdown in the ice sheet retreat rate and a widespread standstill of the SWLIS marked by the formation of the regionally extensive Cree Lake Moraine complex. At the same time, we record a switch from topographically constrained ice streaming in the Hay, Peace, Athabasca, and Churchill river lowlands to a regionally extensive and unconstrained (non-ice streaming) sheet flow that fed the ice margin at its Cree Lake Moraine position. While it is highly likely that these dynamics were influenced by the external climate forcing of the YD, the regions' underlying lithology and topography likely moderated the ice sheets' response to external forcing. Interestingly, these changes in ice flow behaviour and regional ice margin standstill occur at the transition from the soft (deformable) sediment-covered Western Canadian Sedimentary Basin to the predominantly hard (rigid) bed of the crystalline Canadian Shield. While we acknowledge that in some areas, the Canadian Shield is mantled by sediment, the general change in the underlying bed composition (from predominantly soft bedded to predominately hard bedded) likely resulted in an increase in bed strength and frictional resistance at the ice-bed interface, resulting in a decrease in ice velocity and, thus, potentially a slowing of the rate of ice sheet retreat perhaps even independent of external climate forcing (Bradwell et al., 2019). This pattern is analogous to changes observed in the northern portions of the British Irish Ice Sheet (Bradwell et al., 2019; Clark et al., 2022a), where bedrock lithology played a similar role in changing ice sheet dynamics. In the SWLIS, a transition from a relatively soft to a hard bed resulted in a reduction in ice retreat rates and changes in ice stream behaviour. Furthermore, it is plausible that a shift from retreat down a reserve bed slope over the Western Canadian Sedimentary Basin, with large time-transgressive ice marginal glacial lakes (see Fig 6), to retreat across a normal bed slope, with less extensive impounded lakes on the Crystalline Shield also decreased retreat rates (e.g., (Utting and Atkinson, 2019; Stokes and Clark, 2004; Evans et al., 2012). This was due to the changes in water availability at the ice-bed interface and the influence of this on glacier dynamics (Carrivick and Tweed, 2013). The simultaneous change in lithology, bed slope and climate experienced by the SWLIS are in contrast to many other portions of the LIS at this time and may therefore help explain why this sector of the LIS has such a marked response to YD cooling, especially in comparison to other portions of the western Laurentide margin (e.g., Gauthier et al., 2022; Reyes et al., 2022)."_

There may be an issue with figure numbering at the end of the manuscript, it appears figure 7 isn't referenced in the text, though a non-existent figure 6a is referenced.

This has now been amended. Non-existent Figure 6a references have been removed and replaced. Figure 7 is now referenced in the text.

**Line by line notes:**
55. There is inconsistency in "Fig." vs "Fig" in figure references (only an issue when one tries to search where a figure is referenced in the text).

This has now been amended. All figures are referenced as "Fig."

63. I don't think pre is needed here to describe existing chronologies.

'Pre' is removed when describing existing chronologies.

95. It's not clear what "these" refers to in this sentence.

We have removed "these" from the sentence on line 95 and rephrased as follows:

*"While the reconfiguration of these ice streams can be clearly seen in the geomorphic record, they have not been reconstructed in detail for the entire ice sheet sector."*

104. I'm not sure biases is the right word here. Perhaps use lag or lag in the timing to describe the temporal offset between deglaciation and biotic colonisation?

This has now been amended: *"Until recently, the deglacial chronology of the SWLIS was based primarily on radiocarbon data (in compilations by; Dyke, 2004; Dalton et al., 2020), the application of which is limited by the unquantified temporal offset between deglaciation and biotic colonization (c.f. Froese et al., 2019)."*

164. The numbers on the flowsets in the panels of Figure 4 are very difficult to read. This either requires having the google earth document open or a copy of Figure 3 which is more legible.

We agree that the flowset numbers are very small due to the sheer number of flowsets mapped. We do not remove the numbers as we believe they should be labelled within the manuscript. We now include a hyperlink to the .kml file where a reader can search for an individual flowset in google earth in the figure caption.

We also note that some labels are hidden under town or city names. We have shifted these as shown in the screen print below:

[Figure]

250. It would be useful to note here that the large-scale reconstructions you are using (Dyke's and Dalton's) are based on radiocarbon only, while the data set you are using have data from multiple dating techniques. I don't think there is an issue with this approach, given the scale and approach, but I do think the discrepancy should be pointed out to the reader.

Yes, we agree. We use only the geographic position (defined by the geomorphology in the region) defined by Dyke and Dalton and reassign an age to this location based on our new chronology (Norris et al., 2022).

*This has now been clarified in the methods section: "We combine this chronology with the mapped ice margin positions of Dyke (2004) and Dalton et al. (2020), reassigning a new age to each position based on the chronology of Norris et al. (2022), and present recessional isochrones for the SWLIS."*

270. Any hypothesis for where these ice divides might have been or how far away?

Unfortunately, what is preserved in the geomorphic record tells us little about the position of the ice margin. Hence, we make no real inferences about this position.

To make this really clear to the reader, we have slightly amended line 270:

*"Subglacial landforms that comprise flowsets in the Grande Prairie and Calgary regions have been interpreted by Atkinson et al. (2016) to signify synchronous build-up of the CIS and LIS. We support this interpretation and suggest that these flowsets must have been active prior to the formation of an extensive CIS-LIS ice saddle, which would have deflected ice flow to the south."*

We have also added the phrase 'position unknown' to panel 4a for clarity:

[Figure]

327. I'm not sure what is meant by "extensive ice thickness"?

This statement was unclear, and we have now appended the paragraph. Text now reads:

*"In contrast to other sectors of the LIS, LGM ice streaming in the SWLIS flowed unconstrained by topography. This is likely due to large ice thicknesses in the region immediately southwest of the Keewatin Ice Dome, compounded with the lack of significant subglacial relief over much of the area (Ross et al., 2009; Ó Cofaigh et al., 2010; Margold et al., 2015b). Numerical models estimate average LGM ice thicknesses in these regions vary from >1500 to <2000 m (Tarasov et al., 2012; Peltier et al., 2015; Gowan et al., 2016; Lambeck et al., 2017). Similar ice thicknesses are also evidenced by observations of glacial isostatic adjustment and postglacial delevelling of proglacially formed palaeo-lake shorelines (e.g. glacial lakes Agassiz and McConnell) (Breckenridge, 2015)."*

358. Is it possible to quantify these retreat rates? Otherwise, it would be useful to make a comparison here, rapid retreat with respect to something (either a timing or specific region).

Yes. Retreat rates for the SWLIS were estimated by Norris et al., 2022. We have now included these details. Text now reads:

*"Following the initial separation of the LIS and CIS, the SWLIS experienced rapid downwasting and retreat (Norris et al., 2022). Comparing the initial position of multiple south-eastward flowing networks, we agree with the previous suggestions of Ross et al. (2009) and Ó Cofaigh et al. (2010) who propose that these ice streams would have stopped after the development of a deglacial corridor commonly referred to as the 'ice-free corridor', between the CIS-LIS; otherwise, the ice streams would have had insufficient input from the west to drive flow southeastward. Norris et al. (2022) suggest the timing of LIS retreat across the Interior Plains also implies that following initial detachment from the CIS, rapid retreat rates (380-340 m/yr) continued to characterize SWLIS deglaciation, contemporaneous with BA warming. This chronology suggests that southeasterly-oriented ice flows (IS2 complex) would have been very short-lived, operating for as little as ~500 yrs before they stopped and westerly flows became active in their place as the ice margin retreated."*

393. I think this is meant to be "Bordering" rather than "boarding".

Yes! Typo amended.

400. Can you provide any locations here? The references are useful, but place names would help a reader who is unfamiliar with the literature base.

Amended. We have added location descriptions to this section.

Text now reads: *"This is particularly evident in central Alberta, at the termini of fs 100, 138, 147 (northeast of Red Deer) and the region surrounding the Neutral Hills demarcated by fs 33,48 and 60 (Evans et al., 2008; Atkinson et al., 2018; Utting and Atkinson, 2019; Evans et al., 2020)."*

454. Can you provide a bit more glaciologic reasoning for this? (see broader comment above)

Please see our response and changes made above relating to the broader comment.

471. Is this point still true, it sounds like this paper has at least in part clarified this issue.

Yes, this is correct. We have amended the text to read:

*"While switches in ice sheet dynamics and changes in ice-marginal behaviour have previously been recognized for the SWLIS (i.e., Ó Cofaigh et al., 2010; Ross et al., 2009; Margold et al., 2018; Evans et al., 2020), the duration over which these dynamic changes occurred and how they relate to climate forcing is less clear."*

548. There is no Figure 6a, I think this is Figure 7a.

Amended.

644. The supplemental file wasn't able to be opened without adding".kmz" to the file name.

Amended. A new file will be uploaded upon the editor's request of the final manuscript, figures and files.

Figure 4- The numeric labels for the flow sets is illegible at the current size/resolution. Either they can be omitted or should be made larger to be readable.

Please see the comment above relating to the labels.

**Reviewer 2 comments:**
We thank Reviewer 2 for the constructive comments on the manuscript. Below we outline the changes made in reference to the reviewer's text comments.

This is a well-written study that reconciles previously compiled flowsets with new mapping in a vast area of the SW LIS constrained by a recently updated deglacial chronology. The discussion is supported by a well-constructed schematic model and addresses relevant scientific questions within the scope of TC. Overall a very good contribution that warrants publication after minor revisions. I recommend to clarify the interpretations of some flowsets and relative chronology where the data is poorly constrained, and indicate where the new mapping was completed. I also suggest improving several figures where it is laborious to see the flow directions and labels of the flowsets or the contrast in elevation between the uplands and the lowlands, important to appreciate the control (or not) on the ice streams during the various proposed phases. Also I question the use of Dalton's ice margins on Figure 4 since a new deglacial chronology is available by the first author in this region, as well as the parallel flowset pattern over the Canadian Shield where the known record is complex and where ice streams have been suggested across and down-ice of the Athabasca Basin. Below are my detailed comments where suggested edits/revisions are recommended.

**Detailed comments**

- Line 61: Was the mapping entirely done from remote imagery interpretation?

Geomorphic interpretations and new flowset mapping were made using pre-existing geomorphic maps. The majority of this mapping was from remote sensing data, however in some cases regions were ground-truthed. We have amended section 3.1 to reflect this.

Text now reads:

*"In order to provide a complete reconstruction of the ice flow configuration and dynamics as well as the ice-marginal retreat, the glacial geomorphology of the SWLIS was synthesized from pre-existing mapping (Ó Cofaigh et al., 2010; Atkinson et al., 2014; Evans et al., 2016; Norris et al., 2017; Atkinson et al., 2018; Evans et al., 2020). The following landform categories were collated from pre-existing mapping: ice flow parallel lineations (flutings, drumlins, mega-scale glacial lineations (MSGL), and crag-and-tail features), moraines (major and minor), ice-thrust ridges (and glaciotectonic rafts), crevasse fill ridges, meltwater landforms (major and minor meltwater channels and eskers), kames/hummocks, palaeo-shorelines and dunes. Pre-existing mapping comprises features that were predominantly identified from satellite imagery combined with detailed aerial photograph interpretation. In many locations, features have been ground-truthed by the original authors (e.g. Matthews et al. 1975; Atkinson et al 2014, 2018 and references therein). In particular, geomorphic mapping from LiDAR imagery (5-15 m resolution) at a 1:10,000 scale was utilized for Alberta (Atkinson et al., 2014, 2018). For all other parts of the study area, geomorphic mapping was compiled at 1:30,000 (Norris et al., 2017), 1:10,000 (Ó Cofaigh et al., 2010; Evans et al., 2016, 2020) and 1:1,000,000 (Mathews et al., 1975). Due to a lack of accessible LiDAR data within Saskatchewan*

*and northeast British Columbia, mapping has been primarily completed from a combination of 1-arc second SRTM (~30 m resolution) and ALOS (~30 m resolution) imagery and aerial photograph analysis."*

- Lines 140-142: Considering the lack of striations on the Western Canadian Sedimentary Basin, were striae on boulder pavements considered to help define the relative chronology of the flowsets?

While some striae on boulder pavements may have been considered by original authors in their mapping, this is not highlighted in their methods. We based our interpretations of the original geomorphic mapping and previous flowset mapping in the SWLIS region.

- Line 175: Clarify from which ice source is the SE to NW movement (CIS or LIS).

These flowsets were first mapped by Atkinson et al. (2016). As we describe in lines 268-270 these authors assume this ice is of CIS origin. We support this interpretation and add the following details to our results section:

*"Sourced from the Rocky Mountains, southeast to northwest trending ice flow is recorded within the Grande Prairie region (fs 310, 324, 361)."*

- Lines 199-200: Were the hummocky and stagnant ice terrains compiled from your recent mapping? This is the first time these are mentioned in the text. Or refer to previous work.

This was mistakenly omitted from the list of landforms in section 3.1, Geomorphological mapping. The text has now been amended. Text now reads:

*"The following landform categories were collated from pre-existing mapping: ice flow parallel lineations (flutings, drumlins, mega-scale glacial lineations (MSGL), and crag-and-tail features), moraines (major and minor), ice-thrust ridges (and glaciotectonic rafts), crevasse fill ridges, meltwater landforms (major and minor meltwater channels and eskers), kames/hummocks, palaeo-shorelines and dunes."*

-Line 205: Indicate the location of the Churchill River lowlands on one of the figures.

Amended.

- Line 229-230: Indicate the possibility of the SE flow in the Churchill River lowlands to be part of Phase 3 (if not crosscut by other flowsets).

These flowsets are crosscut (fs 66 and 25) the processive migration of SE flow that is part of IS 4/5 is also marked by the crosscut shear moraines depicted in Phase 4.

- Line 231: Flow directions ("from southwest-west to the northeast, northwest-to-southeast and north-to-southeast flow") are incomprehensible. Revise.

Agreed- this was a needlessly complex way of phasing this. We have amended the text as follows:

*"Associated with fs 150, two cross-cutting lateral shear moraines document a time-transgressive switch from northeastward ice flow that migrates over time in a clockwise direction with the youngest flow depicting southeastward ice flow. The cross-cutting relationships exhibited by these landforms suggest they formed as a shifting ice drainage system migrating from a west to north source as the ice sheet thinned (i.e., from Phase 3 to 4)."*

- Line 240: There are many streamlined sediment landforms on the Shield, in addition to lineated outcrops, particularly over the Athabasca sedimentary Basin (e.g. Schreiner, 1984; Campbell, 2007, 2009; Campbell et al., 2007).

We agree that there are many regions of lineated soft sediment. We now amend our statement in section 4.5 Phase 5, to include this information. Text now reads:

*"Phase 5 documents the last stages of ice flow to the southwest (Fig. 4e). During this phase, ice flow terminated at the Cree Lake Moraine and Upper Cree Lake Moraine complex. A large regional-scale flowset (fs 164) and a series of smaller retreat and localized readvance flowsets (fs 65, 77, 94, 101, 114, 118, 135, 140 and 161) comprise lineated outcrops of the Canadian Shield and streamlined sediments."*

- Line 243: Could some of the landforms on the uplands be older and preserved under cold-based thinner ice?

We initially thought this might be the case, and indeed on several uplands, fragments of older ice flow do exist and are documented in Phase 1. However, the flowsets we describe in this section on the uplands are small discrete flowsets that are associated with ice-marginal features (moraines, meltwater channels). These landforms also cross-cut recessional moraines and glaciolacustrine sediments within the surrounding lowlands, suggesting that the lowland regions were ice-free and that proglacial lakes in the area had drained.

- Line 263: Consider if the Cameron Hills fragments could have been part of Phase 4 to the SW.

These flowsets have been previously interpreted by Margold et al. (2015a and b) as being the oldest in the Cameron Hills region. These flowsets are also crosscut by flowset 398.

- Line 269: Ice divide or ice dome? Is there any evidence for an ice dome within your study area during LGM? Refer to previous work if necessary.

We have reworded this sentence as it was misleading. We do not have evidence of an ice dome in our study area during the pre-LGM. We were instead referring to the CIS-LIS ice saddle buildup. This is now better clarified as follows:

*"We support this interpretation and suggest that these flowsets must have been active prior to the formation of an extensive CIS-LIS ice saddle, that would have deflected ice flow to the south."*

- Line 317: If there is no convergent flow in the CAIS/HPIS, why do you show a converging onset zone up-ice of the multiple flowsets on Fig. 4b (in light grey over the Shield boundary)?

We have amended the shape of our convergent flowlines in Figure 4b. The figure now depicts flow as:

[Figure]

- Line 363: Phase 3 is discussed in this section (with multiple references to Fig. 4c). Add Phase 3 in subtitle or in title of section 5.4.

The references to Fig 4c should have been to Figure 4d. We have now amended these.

- Lines 412-413: Indicate where the Peace River lowlands are on a figure; refer to Fig. 4d at end of sentence.

Figure 4d is now referenced and the Peace River lowlands are labelled on this figure.

- Line 430: Indicate on a figure where the spillway is.

The location of the Clearwater Lower Athabasca Spillway is now shown in the figure.

[Figure]

- Line 449: Refer to Fig. 4d.

Reference to Fig. 4d added.

- Line 452-454: There are also some Devonian lithologies (harder) at the Shield boundary. On the other hand, the Athabasca Basin sandstones form a large part of the Shield under your "Cree Lake ice flow" – with much softer rocks and an extensive sediment cover, up to 100 m+. So not entirely hard-bedded conditions over the Shield. Consider how this may affect your interpretations of a subglacial lithological control at the Shield boundary.

We acknowledge that portions of the Canadian Shield are indeed covered by packages of soft sediment, our discussion aims to focus on the broad-scale transition that is taking place. We have clarified and included a statement in our discussion to highlight this to the reader.

Amended text now reads:

*"While we acknowledge that in some areas, the Canadian Shield is mantled by sediment, the general change in the underlying bed composition (from predominantly soft bedded to predominately hard bedded) likely resulted in an increase in bed strength and frictional resistance at the ice-bed interface, resulting in a decrease in ice velocity and, thus, potentially a slowing of the rate of ice sheet retreat perhaps even independent of external climate forcing (Bradwell et al., 2019)."*

- Line 551: Wrong reference to Fig. 3?

We have now amended it to reference Fig. 5.

- Line 570: Refer to Fig. 7b?

Reference to Figure added.

- Line 576: Refer to Fig. 7c?

Reference to Figure added.

- Line 588: Refer to Fig. 7d?

Reference to Figure added.

- Line 600: Perhaps a good indicator of terrestrial ice sheet collapse but mainly pertinent to soft-bedded areas?

We agree, our interpretation of the geomorphology of the SWLIS is an example of terrestrial ice sheet collapse in regions with ample deformable sediment. W amend the text to reflect this:

*"Although the response of the SWLIS to climatic change is complex, we suggest that its geomorphic imprint identified here is not unique and propose that the combination of regions of rapid ice marginal retreat and ice sheet disintegration and downwasting may be a valuable indicator of terrestrial ice sheet collapse, especially in soft-bedded regions."*

**Suggested edits on Figures**

Fig. 1: Add description of inset map and source; it is very difficult to see the uplands/valleys and appreciate control (or not) on ice streams on main figure:  I suggest enhancing DEM colors and/or using hillshades (same comment applies for Figs 3, 4, 5 and 6).

The hillshade and ice sheet transparency has been changed for all figures. An example for figure 1 is shown below:

[Figure]

The figure caption is amended to include a reference for the LGM extent of the North American Ice Sheet (Dalton et al., 2020):

*"Figure 1: Regional ice-flow imprint of the southwestern sector of the Laurentide Ice Sheet. Digital elevation model (DEM) from the Shuttle Radar Topographic Mission (SRTM) provided by the USGS EROS Archive (EROS Centre, 2018). Last glacial maximum ice extent from Dalton et al. (2020)."*

Fig. 2c: It is too small to see anything; also, it does not correspond to same area as a-b-d-e (but it does to outline on Fig. 1).

We enlarge the size of panel c and overlay it on a portion of the extent shown in panel b. A screenshot is shown below:

[Figure]

Fig. 3: Which flowsets are new from this study? I recommend adding an inset map showing where the main areas of changes/additions are located; it is tedious to figure out the directions (arrows) of the ice flow within the flowsets: I suggest making lines thinner inside the flowset;

We have added a new figure panel. Figure 3a. This now shows the location of the previous flowset mapping:

[Figure]

We also amended our methods text to highlight the work of previous studies in mapping flowsets.

Amended text now reads: *"We synthesized and updated flowsets (mapped and originally presented by Evans et al., 2014; Atkinson et al., 2016; Utting et al., 2016; Norris, 2019; Atkinson and Utting, 2021) and generated new flowset mapping in regions where no mapping was available (~25% of the flowsets) (Fig. 3a). In addition to mapped glacial geomorphology, in many cases, previous regional scale delineation of ice flow corridors or ice stream locations guided our flowset identification (Evans, 2000; Ross et al., 2009; Evans et al., 2008; Ó Cofaigh et al., 2010; Evans et al., 2012; Margold et al., 2015b, a; Stokes et al., 2016; Margold et al., 2018; Evans et al., 2020) (Fig. 3a). Each flowset was then divided into one of six categories: (i) ice stream, (ii) event, (iii) deglacial retreat, (iv) deglacial readvance, (v) palimpsest and (vi) unknown (Table 1). Flowsets were also ordered according to their relative age based on overprinting and cross-cutting relations (Kleman et al., 2006; Greenwood and Clark, 2009; Kleman et al., 2010; Hughes et al., 2014) to reconstruct changes in ice sheet dynamics over space and time."*

We have amended the line of all flowsets in Fig. 3b

[Figure]

Previous mapping in the NE (area of parallel flowset #164) supported by field datasets shows much more complex ice flows, with some diverging and many x-cutting, and probable ice streams (soft-bedded and hard-bedded). Also the Maskwa system was depicted as reaching much further up-ice into the ice sheet by Ross et al (900 km long including 350 km on the Canadian Shield) based on mapping and till composition. Mention alternative interpretations to what you are proposing here for the Shield.

For this region of the map, we compile geomorphic mapping from Norris et al. (2017) and flowset mapping from Norris (2019). We also note in this region we have 6 flowsets that crosscut the fs164, however we acknowledge that there might be complexity that is not visible from SRTM based mapping. We amend the position of the Maskwa Ice Stream to extend ~350 km onto the Canadian Shield in line with mapping and till geochemistry information described by Ross et al (2009).

Fig. 4:  Add Phase # on each figure 4a,b,c,d,e; the ice margins as depicted appear to be those of Dalton et al? It is confusing since you constrain the ice retreat using your updated chronology. Can you use your modelled ice margin retreat from the 2022 paper? Add the ages on each isochrone; it is very difficult to see the numbers, writings, and different colors representing ages of ice flows - revise; add a scale on Figs. 4b-e.

Amended. Phases and scale bars have now been added to figure 4. We use the position of the isochron drawn by Dalton et al 2020 and assign a new age to the position based on the chronology of Norris et al 2022. Given the uncertainty in the compiled 14C, TCN and OSL ages described in Norris et al 2022 (and briefly redescribed in this manuscript) we do not assign a single age to any isochron in an attempt to not overinterpret our data. Instead, we add an age range to each panel and highlight key chronologically constrained points in text.

[Figure]

[Figure]

Fig. 4a: The generalized ice flow legend symbol is too small and very difficult to follow on each map. Enhance colors and increase size.

We have amended the line of all flowsets in Fig 4 as we show above to match the thinner line thickness as in Fig. 3.

Fig. 4b: Add names/numbers of the three corridors; what are the grey polygons surrounding the flowsets?; is the CIS/LIS saddle location well defined?

Names and locations of the three fast flow corridors have been added (Maskwa, Central Alberta IS and, High Plains IS).

The grey outlines depict the generalised ice flow (gradient of colours depicts oldest to youngest flows in a single phase.

[Figure]

[Figure]

Fig. 4c: Show or number the six NW to SE ice flow corridors.

[Figure]

Fig. 4d: One flowset within IS 4/5 appears in Phase 2 as well.

Flowset 49 has been removed from Panel b (Phase 2).

**Editorial team comments:**

For the next revision, please check if your figures containing maps/aerial images require a copyright statement/image credit and add it to the figures (or captions) (https://publications.copernicus.org/for_authors/manuscript_preparation.html#mapsaerials). If these figures were entirely created by the authors, there is no need to add a copyright statement or credit. In that case it is important that you confirm this explicitly by email.

*Amended. Text now reads:*

*"Figure 2: Regional ice-flow imprint of the southwestern sector of the Laurentide Ice Sheet. Digital elevation model (DEM) from the Shuttle Radar Topographic Mission (SRTM) provided by the USGS EROS Archive (EROS Centre, 2018). Last glacial maximum ice extent from Dalton et al. (2020).*

*Figure 2: Classification of glacial geomorphology into flowsets. (a) Shuttle Radar Topography Mission (SRTM) imagery displaying the onset zone of the Central Alberta Ice Stream. (b) Area of corresponding geomorphological mapping (from Atkinson et al., 2018) and, (c) surficial geological units (from Fenton et al., 2013)(d). Simplification of geomorphic mapping and surficial geological units into 'flowlines' (e) Classification of flowlines into individual*

*flowsets as either 'ice stream', 'deglacial retreat', "deglacial readvance' 'event' 'palimpsest' or 'unknown. See Figure 1 for the location of Figure 2 panel. Digital elevation model (DEM) from the Shuttle Radar Topographic Mission (SRTM) provided by the USGS EROS Archive (EROS Centre, 2018).*

*Figure 3: Flowset map of the southwestern Laurentide Ice Sheet (available as a supplementary .kml file). Flowset configuration reflects a highly dynamic system of fast flow caused by multiple reconfigurations in ice sheet drainage. Digital elevation model (DEM) from the Shuttle Radar Topographic Mission (SRTM) provided by the USGS EROS Archive (EROS Centre, 2018). Last glacial maximum ice extent from Dalton et al. (2020).*

*Figure 4: Deglacial ice flow dynamics of the southwestern Laurentide Ice Sheet. This reconstruction presents a comprehensive synthesis, collating pre-existing ice stream mapping with ice streams presented in this study.  Timing of ice retreat and associated ice flow reorganizations are constrained using the deglacial chronology of Norris et al. (2022). Digital elevation model (DEM) from the Shuttle Radar Topographic Mission (SRTM) provided by the USGS EROS Archive (EROS Centre, 2018). Last glacial maximum ice extent from Dalton et al. (2020).*

*Figure 5: Regional geomorphic imprint of the distribution of ice sheet active retreat and punctuated stagnation across the southwestern Laurentide Ice Sheet. Digital elevation model (DEM) from the Shuttle Radar Topographic Mission (SRTM) provided by the USGS EROS Archive (EROS Centre, 2018). Last glacial maximum ice extent from Dalton et al. (2020).*

*Figure 6: Distribution of reconstructed proglacial lakes associated with the retreat of the southwestern Laurentide Ice Sheet, classified by relative age. Digital elevation model (DEM) from the Shuttle Radar Topographic Mission (SRTM) provided by the USGS EROS Archive (EROS Centre, 2018)."*